# Systematic Review of Binge Eating Rodent Models for Developing Novel or Repurposing Existing Pharmacotherapies

**DOI:** 10.3390/biom13050742

**Published:** 2023-04-25

**Authors:** Gregory Berger, Joshua D. Corris, Spencer E. Fields, Lihong Hao, Lori L. Scarpa, Nicholas T. Bello

**Affiliations:** 1Endocrinology and Animal Biosciences Graduate Program, Department of Animal Sciences, School of Environmental and Biological Sciences, Rutgers, The State University of New Jersey, New Brunswick, NJ 08901, USA; 2Department of Animal Sciences, School of Environmental and Biological Sciences, Rutgers, The State University of New Jersey, New Brunswick, NJ 08901, USA; 3Nutritional Sciences Graduate Program, School of Environmental and Biological Sciences, Rutgers, The State University of New Jersey, New Brunswick, NJ 08901, USA; 4Rutgers Brain Health Institute, Rutgers University and Rutgers Biomedical and Health Sciences, Piscataway, NJ 08854, USA

**Keywords:** eating pathologies, binge, Vyvanse, fluoxetine, Prozac, rat, mouse, bulimia nervosa

## Abstract

Recent advances in developing and screening candidate pharmacotherapies for psychiatric disorders have depended on rodent models. Eating disorders are a set of psychiatric disorders that have traditionally relied on behavioral therapies for effective long-term treatment. However, the clinical use of Lisdexamfatamine for binge eating disorder (BED) has furthered the notion of using pharmacotherapies for treating binge eating pathologies. While there are several binge eating rodent models, there is not a consensus on how to define pharmacological effectiveness within these models. Our purpose is to provide an overview of the potential pharmacotherapies or compounds tested in established rodent models of binge eating behavior. These findings will help provide guidance for determining pharmacological effectiveness for potential novel or repurposed pharmacotherapies.

## 1. Introduction

Advancements in uncovering novel therapeutics for eating disorders have been limited in the past several decades. Treatments for eating disorders have historically depended on cognitive behavioral and psychotherapies [1,2,3]. While these therapies remain effective for specific eating disorders and coping with individualized factors, the long-term effectiveness of these therapies is dependent on the availability of trained clinicians, patient commitment, and other clinically managed outcomes [4,5,6]. In recent years, there has been an interest in developing pharmacotherapies as adjuvant to improve behavioral and psychotherapy outcomes [7,8]. In the US, there are only two FDA-approved pharmacotherapies options for the treatment of eating disorders. Both medications are “repurposed” and were originally approved for other psychiatric illnesses [9]. Fluoxetine (FLX) is available for the clinical management of bulimia nervosa (BN), whereas lisdexamfetamine (LDX) is available for binge eating disorder (BED). To date, there is no pharmacotherapy approved for use in patients with anorexia nervosa (AN). Notably, BN and BED are two separate psychiatric disorders with binge eating as a core pathology but differ in their clinical diagnostic criteria and management [10].

FLX was FDA-approved in 1987 for the treatment of major depressive disorder and then approved for acute management and maintenance of BN in 1994 [11,12]. Efficacy of FLX was determined in two pivotal randomized placebo-controlled multi-center studies [13,14]. In an 8-week study (*n* = 387, 100% were female; DSM III-R defined BN), FLX (60 mg/day) had a 20% placebo-subtracted reduction of binge eating and a 31% placebo-subtracted reduction of vomiting episodes per week. Discontinuation rates were 37.2% for placebo and 30.2% for FLX [13]. Similar reductions in bulimic behaviors were demonstrated in a 16-week study (*n* = 398, 96.2% were female; DSM III-R defined BN) [14]. FLX (60 mg/d) had a 32% placebo-subtracted reduction in binge eating and a 29% placebo-reduction reduction of vomiting episodes per week. Discontinuation rates were 52% for placebo and 40.5% for FLX [14]. Recent evidence suggest that FLX is still considered effective for the management of BN, but approximately 60% of BN patients do not achieve remission with FLX or available psychotherapies or their combination [15].

The other available pharmacotherapy is LDX, which was initially FDA-approved for the management of attention deficit hyperactivity disorder (ADHD) in 2007 for children (6–12 years old) and 2008 for adults. In 2015, LDX was FDA-approved for the maintenance of moderate to severe BED [16]. Several randomized placebo-controlled phase III studies were conducted to determine the efficacy of LDX for BED. Two 12-week studies (Study 1: *n* = 379, 86.5% were female; Study 2: *n* = 366, 85.2% were female; DSM-IV TR defined BED; NCT01718483 and NCT01718509) demonstrated LDX (50 or 70 mg/day) had a 26.3% placebo-subtracted reduction (study 1) and a 37.6% placebo-subtracted reduction (study 2) in binge eating days/per week [17]. Discontinuation rates were 18% and 25.9% for placebo and 17.7% and 26.5% for LDX for each study, respectively [17]. Efficacy of LDX was further established in a 26-week study (*n* = 418, 87.1% female; DSM-IV TR defined BED; NCT02009163), which demonstrated a reduced relapse risk for BED patients [18]. LDX-treated subjects had a 3.7% relapse rate compared with 32.1% relapse rate in placebo-treated subjects [18]. LDX is also likely to improve the high impulsivity reported in some BED populations [19]. Using a within-subjects design, it was demonstrated that 8-week LDX (50 mg/day or 70 mg/day) decreased binge eating frequency in BED subjects (*n* = 41, 97.6% female; DSM-V defined BED) [19]. Notably, binge episode frequency was also quantified as *binge days* per week rather than discrete binge episodes, since self-reporting of when binge episodes start and finish can be difficult to delineate or recall [17]. Additional medications initially approved to treat other psychiatric illnesses have been investigated in BED or BN populations [20,21,22,23,24] and are reviewed in detail elsewhere [25,26].

Concerns for the safety and tolerability of pharmacotherapies for BN and BED arise on several levels. The frequency and severity of treatment emergent adverse events represents a major contributor to the discontinuation rate in efficacy trials [14,27]. Pregnancy risks are major concerns for women of child-bearing age, which represents the predominant age range of women seeking therapeutic assistance for eating disorders. For instance, LDX and FLX has not been systematically tested for their effects on a developing fetus, but there is ample evidence in rodent and non-rodent species to suggest a pregnancy risk (FDA Pregnancy category C) [12,28]. LDX also has an abuse and dependency potential with a long-term risk to elevate cardiovascular endpoints [29]. For the most part, though, FLX and LDX are considered to be well-tolerated medications for the treatment of BN and BED, respectively. 

Unlikely other classes of medications [30], to date, there is no guidance for industry on developing pharmacotherapies for the clinical management of binge eating. However, in the FDA application for LDX for moderate to severe BED, there was a stated assertion by the sponsor that a mean reduction of 0.5 or more binge days per week was “clinically meaningful” [31]. Additional endpoints include number of subjects with total remission of binge eating and/or vomiting, improvements in structured diagnostic assessments (i.e., Y-BOCS-BE, BES, EDI, CGI, BDI, etc.), and triglyceride levels in BED patients [13,17,28,31]. As such, the primary endpoint for determining efficacy of a pharmacotherapy for BED or BN is a sustained significant reduction in the *frequency* of binge (BN/BED) or vomiting (BN) episodes [13,20]. A *reduction in binge frequency* outcomes could also represent only one component in multi-stage optimization innervation [32]. Nonetheless, *reduction in binge frequency* represents a measurable behavioral endpoint for determining pharmacotherapeutic effectiveness. 

Animal models, specifically rodent models, are used to advance the screening and development of potential pharmacotherapies for psychiatric disorders [33,34]. In addition, animal models can assist in the early identification of some safety and tolerability-related issues. No animal model, however, can fully encompass all the intricacies of a psychiatric disorder. To be relevant and effective, animal models should be developed to ask specific hypothesis-testable questions for particular features of the complex psychiatric disease under investigation [35]. Within the field of eating disorders, binge eating has several core components that can be recapitulated in rodents for assessing the effectiveness of potential pharmacotherapies. While the sense of ‘loss of control’ of binge eating is a pathological feature that is difficult to objectively assess in animals, two aspects of binge eating that can be recapitulated are *excessive ingestion of calories in a short period of time* and *the episodic pattern* of eating. In addition, some rodent binge eating models have incorporated a *caloric restriction* component to mimic restrictive eating or cycling dieting influencing binge eating behaviors [10]. Rats and mice lack an emetic response [36], so calorie restriction in animal models also serves as a method of assessing the influence of calorie restrictive conditions on eating without vomiting. The purpose of this systematic review is not to further debate on which experimentally controlled factors appropriately recapitulate the core features of a clinical eating disorder. The purpose of this systematic review is to provide an overview of the potential pharmacotherapies or compounds tested in established rodent models of binge eating behavior. This information can be used to develop hypothesis-testable questions on pharmacotherapeutic options, standardized approaches to assess candidate drugs, and to provide guidance on future experimental design approaches building on existing findings in binge eating rodent models. 

## 2. Materials and Methods

This systematic review was performed using Pubmed (https://pubmed.ncbi.nlm.nih.gov/; accessed 31 December 2022) and APA PsycInfo (https://psycnet.apa.org/search) web-based databases (accessed 31 December 2022). The Preferred Reporting Items for Systematic Reviews and Meta-Analyses guidelines (PRISMA) flowchart was used in the literature selections [37]. The following terms were used: “binge eating”, “binge eating disorder”, “bulimia nervosa”, “animal models”, “rodents”, “rat”, “mouse”, “intermittent”, “intermittent access”, “intermittent palatable food”, “dietary binge eating”, “stress eating”, “stress overeating”, “stress-induced binge eating”, “binge-like eating”, “binge-type eating”, and “binge feeding”. The searches were performed to include all studies with a full publication date before 1 January 2023 (not epub date). Inclusion criteria were studies published as peer-reviewed articles, written in the English language, and definition of the feeding schedule as “binge” or “binge-like” or “binge-type”. Articles were further selected based on pharmacology endpoints to included compounds, agents, drugs, or medication interventions on feeding outcomes. Exclusion criteria were studies that used liquid (i.e., Ensure^®^ or chocolate drink) or sweetened solutions as binge foods. Our rationale for excluding liquid binge food studies was that while these represent related ingestive behaviors and brain processes, fluid intake is also regulated by osmotic and volumetric factors. These factors are distinguishable from eating. Additional studies were excluded that had alcohol- and/or drugs of abuse-related outcomes. Studies were also excluded if the binge paradigm with regards to binge food composition, access, frequency, or duration were poorly defined.

## 3. Results

### 3.1. Study Charactersitics and Selection

As detailed in Figure 1, there were a total of 434 studies screened, and 67 studies were used for systematic review. Studies were further classified based on targeted systems, combinations, or comparisons between compounds. These studies are summarized in Table 1, Table 2, Table 3, Table 4 and Table 5. Given the differences in study design, approach, binge food, and major pharmacological outcomes, the summary of these studies is best represented in a table format.

### 3.2. Biogenic Amines and Related Compounds

As described in Table 1, 16 studies [38,39,40,41,42,43,44,45,46,47,48,49,50,51,52,53] were identified that used compounds acting on the biogenic amine systems.

**Table 1 biomolecules-13-00742-t001:** Biogenic amines and related compounds. Studies using adrenergic/noradrenergic, dopaminergic, serotonergic, GABA, and glutamatergic compounds.

**Adrenergic/Noradrenergic**			
**Study**	**Methods**	**Details**	**Major Findings**
Hicks et al., 2020 [50]	**Species:** Rat **Binge Food:** Chocolate-flavored high-sucrose (50% kcal) AIN-76A-based pellets **Access:** 1 h **Frequency:** Daily **Duration:** >12 days **Calorie Restriction:** None	Male Wistar rats were training to nose poke for binge food or standard chow pellets on fixed ratio and progressive ratio (PR) testing. The dose-dependent effects of prazosin (0.5–2 mg/kg; IP) on binge food responding were assessed.	Alpha adrenergic receptor 1 antagonist, prazosin, resulted in reduced food responses, but increased the PR breakpoint. Increased PR breakpoint were observed at all doses tested in rats receiving the binge food, but only at the 2 mg/kg in the rats receiving standard chow.
Bello et al., 2014 [43]	**Species:** Rat **Binge Food:** Vegetable shortening blended with 10% sucrose **Access:** 30 min **Frequency:** 2 non-consecutive days per week **Duration:** 6 weeks **Calorie Restriction:** Two groups of rats had a 24-h calorie restriction prior to the binge period twice a week (Restrict).	Female Sprague Dawley rats were exposed to the binge eating paradigm with or without intermittent calorie restriction. c-Fos immunoreactivity was measured in the medial prefrontal cortex (mPFC) and hypothalamic paraventricular nucleus (PVN) in response to binge food, with or without acute restraint stress (1 h). In another set of experiments, female rats were implanted with an osmotic minipump to provide a sustained release of guanfacine, alpha adrenergic 2A receptor agonist (0.5 mg/kg/day) or vehicle to determine the effects of binge food intake.	Binge intake was similar with or without intermittent calorie restriction (Restrict). Acute restraint stress-induced c-Fos was higher in the mPFC in the rats with binge food access. Acute stress increased total c-Fos in the PVN. Chronic guanfacine increase binge food intake in the Binge group, but not Binge + Restrict.
Bello et al., 2014 [44]	**Species:** Rat **Binge Food:** Vegetable shortening blended with 10% sucrose **Access:** 30 min **Frequency:** 2 non-consecutive days per week **Duration:** 10 weeks **Calorie Restriction:** Two groups had a 24 h calorie restriction prior to the binge period twice a week. (Restrict)	Male Sprague Dawley rats were exposed to the binge eating paradigm with or without intermittent calorie restriction. Noradrenergic involvement was assessed by feeding suppression to a selective norepinephrine reuptake inhibitor, nisoxetine (3 mg/kg; IP), dorsal noradrenergic bundle lesions with neurotoxin (DSP4), and single-unit electrophysiological recording of locus coeruleus (LC) neurons.	Nisoxetine-induced feeding suppression was diminished in the Restrict + Binge group. Restrict + Binge group also had a reduced latency to feed after calorie restriction (48 h) and novel environment. Total binge intake 3 weeks after DSP-4 lesion was reduced in Restrict + Binge group. Sensory-evoked LC neural responses were reduced in Restrict + Binge group.
**Dopaminergic**			
**Study**	**Methods**	**Details**	**Major Findings**
Presby et al., 2020 [51]	**Species:** Rat **Binge Food:** Ground Cadbury’s Dairy Milk Chocolate **Access:** 2 h **Frequency:** 12 non-consecutive days (1–3 days apart) **Duration:** 4 weeks **Calorie Restriction:** 85% of ad libitum feeding weights for 18 weeks. After that, 18 weeks of ad libitum feeding and normal weight before starting the binge feeding protocol.	Female Wistar rats were exposed to standard lab chow and divided into three groups, +chocolate, +standard chow, and +empty feeding dish. Effects of lisdexamfetamine (LDX; 0.185–1.5 mg/kg; IP) on operant responding and intake were measured.	No differences between the +standard chow and +empty feeding dish groups. Data were combined to control group. There were dose-dependent effects of LDX on operant responding, but no group effects. Standard chow was reduced at 0.75 mg/kg and 1.5 mg/kg dose in the +chocolate groups. Standard chow was reduced at 1.5 mg/kg in the control group. Chocolate was reduced at 1.5 mg/kg dose in the +chocolate group.
Vickers et al., 2017 [47]	**Species:** Rat **Binge Food:** Ground chocolate (Cadbury’s Dairy Milk) **Access:** 2 h **Frequency:** Intermittent on Days 1, 2, 4, 6, 7, 9, 12, 14, 15, 18, 23, 25 and 28. **Duration:** >28 days **Calorie Restriction:** None	Female Wistar were trained on operant delay discounting lever pressing task (immediate vs. delayed reward) before exposure to the binge eating paradigm. Rats were dosed in a cross-over design with oral lisdexamfetamine (LDX; 0.3 mg/kg or 0.8 mg/kg) or vehicle after 2 weeks on the binge eating paradigm.	Binge eating rats demonstrated responding for the delayed reward. LDX (0.8 mg/kg) normalized binge eating-induced delay discount to non-binge control levels.
Heal et al., 2016 [46]	**Species:** Rat **Binge Food:** Ground chocolate (Cadbury’s Dairy Milk) **Access:** 2 h **Frequency:** Intermittent on Days 1, 2, 4, 6, 7, 9, 12, 14, 15, 18, 23, 25 and 28. **Duration:** >28 days **Calorie Restriction:** None	Female Wistar were exposed to the binge eating paradigm or no access to binge food (control) and trained on conditioned avoidance response task. Binge or control rats were dosed with lisdexamfetamine (LDX: 0.8 mg/kg) or vehicle to assess performance in novel food reward/punishment conflict paradigm.	Binge eating rats had increased time spent in chocolate–pair chambers. LDX (0.8 mg/kg) normalized binge eating-induced percent avoidance and percent escapes to control levels. LDX did not influence avoidance behaviors in control rats
Halpern et al., 2013 [42]	**Species:** Mouse **Binge Food:** High-fat diet (HFD; 60% fat, 20% carbohydrates, 20% protein) **Access:** 1 h **Frequency:** Daily **Duration:** 8 days (3 days; 5 treatment days) **Calorie Restriction:** None	Male C57BL/6J mice were implanted unilaterally into the left nucleus accumbens shell (NAS), core or dorsal striatum with a bipolar tungsten electrode. Mice were exposed to the binge eating paradigm and deep brain stimulation (DBS) with 150 µA on two non-consecutive days. Pharmacological blockage of DBS with D1R antagonist (SCH-23390, 0.075 mg/kg; IP) or D2 antagonist (raclopride; 3 mg/kg; IP) was also performed.	DBS in the NAS reduced binge food intake. Reductions in the NAS DBS suppression of binge eating was observed with raclopride pre-treatment. No differences between SCH23390 and vehicle. DBS of dorsal striatum or core did not influence binge food intake.
**Serotonergic**			
**Study**	**Methods**	**Details**	**Major Findings**
Karth et al., 2022 [53]	**Species:** Mouse **Binge Food:** High-fat diet (39.7% Fat) **Access:** 24 h **Frequency:** Once per week **Duration:** >3 weeks **Calorie Restriction:** None	Male and female mice homozygous for a knockin (KI) of tryptophan hydroxylase 2 (*Tph2*; R439H) or littermate contols (WT). *Tph2* KI mice have reduced (~60–80%) serotonin levels. FLX was administered acutely (30 mg/kg; or saline; IP) or chronically (150 mg/L or unadulterated drinking water) for 21 days. Serotonin-related gene expression was measured by qPCR from the raphe nucleus.	Acute FLX reduced binge food intake in males and females. Overall, male Tph2 KI mice had greater binge food intake, regardless of acute FLX or vehicle treatments. Overall, chronic FLX reduced binge intake. Tph2 KI mice had less FLX-induced binge food intake suppression compared with WT controls. No significant sex by chronic FLX by genotype effects were observed on gene expression. Chronic FLX reduced 5HTR1a in females.
Blumenthal and Pratt, 2018 [48]	**Species:** Rat **Binge Food:** Vegetable shortening **Access:** 2 h **Frequency:** Daily **Duration:** ~22 days **Calorie Restriction:** None	Male Sprague Dawley rats were implanted with bilateral intracranial cannulas targeting the nucleus accumbens core. Rats received vehicle or DAMGO (mu-opioid receptor agonist; 0.025 μg) prior to the binge access. In addition, rats received fenfluramine (0, 0.6, or 3.0 mg/kg; IP) or lorcaserin (0, 0.3, or 1.0 mg/kg; IP).	Fenfluramine dose-dependently reduced binge intake in intra-accumbens vehicle or DAMGO conditions. Lorcaserin only reduced binge intake during the intra-accumbens DAMGO condition.
Price et al., 2018 [49]	**Species:** Rat **Binge Food:** High-fat diet (45%) **Access:** 2 h **Frequency:** Once per week **Duration:** 5–9 weeks **Calorie Restriction:** None	Adult male Sprague Dawley rats were IP administered a novel 5HT-2cR agonist, WAY163909, 0.5, 1.0 or 2.0 mg/Kg) prior to binge access.	WAY163909 reduced binge intake at 1 and 2 mg/kg. Non-binge intake of standard chow was not influenced over the range (0–2 mg/kg) of WAY163909 tested.
Chandler-Laney et al., 2007 [41]	**Species:** Rat **Binge food:** Double-Stuff Oreo cookies **Access:** 3 cycles of restriction–refeeding, each lasting 12 days. Refeeding consisted of presentation of cookies for a short period, followed by ad lib chow. **Frequency:** 12-day cycle: 5 days ad lib chow or restriction, 2 days ad lib chow and cookies, 4 days ad lib chow, 1 day stress/no stress. **Duration:** 6 cycles of 12 days of restriction/refeeding followed by a test day on day 13. **Calorie Restriction:** 66% of ad libitum standard chow. **Stress:** Acute foot shock.	Female Sprague Dawley rats were exposed to a binge paradigm with or without stress or restriction. The central effects of FLX (3 mg/kg;IP) on binge eating were asseseed. Levels of serotonin, dopamine, and metabolites were evaluated from key brain regions involved in feeding and reward. Serotonin, dopamine, norepinephrine, and their metabolites were measured using high-performance liquid chromatography (HPLC).	3 mg/kg dose of FLX suppressed intake of rats with a history of caloric restriction, normalizing the intake of calorie restricted/stressed rats, but had no effect on groups without a history of caloric restriction. Neurochemical differences were not evident in the hypothalamus or tegmentum across groups.
Placidi et al., 2004 [38]	**Species:** Rat **Binge food:** Nabisco Oreo cookies (4.7 kcal/g; 57%, 40%, and 3% of kcal from carbohydrates, fats, and protein, respectively. **Access:** 24 h **Frequency:** After every R-R/S cycle the animals had access to binge food and standard chow. Ad libitum access during test period. **Duration:** ~2 months **Calorie Restriction:** 4 days of 66% of control food intake followed by 6 days of free feeding (R + NS and R + S groups).	Female Sprague Dawley rats were exposed to binge paradigm with or without stress (i.e., acute foot shock; 0.6 mA). The role of serotonergic function in binge eating behaviors was investigated, specifically the influence of FLX (2 mg/kg; IP).	History of caloric restriction alone produced binge-like eating FLX produced binge intake, feeding suppression in rats with a history of caloric restriction and acute stress exposure. The FLX binge intake suppression was abolished when rats were undergoing (i.e., active vs. history) the caloric restriction and had lower body weights.
**GABAergic**			
**Study**	**Methods**	**Details**	**Major Findings**
Wojnicki et al., 2006 [40]	**Species:** Rat **Binge food:** Crisco All-Vegetable shortening **Access:** 1 h **Frequency:** Daily (D) or intermittent (INT) 3 days/week. **Duration:** 4–6 months **Calorie Restriction:** None	Operant performance of 18 non-food deprived male Sprague Dawley rats was assessed under progressive ratio (PR) and concurrent PR-fixed ratio schedules of chow pellets and/or vegetable shortening reinforcement. Rats were then divided into 2 groups based on schedule of binge food access: High-Restriction binge group (1 h access 3 days/week) and Low-Restriction binge group (1 h access daily). The effects of (R)-baclofen (0.3, 0.6, 1.0 and 1.8 mg/kg) on operant performance were assessed.	Increase in lever pressing for binge food was observed in the INT access compared with D access of rats. Pellet responding under the concurrent schedules increased for both groups. For both groups, 1.0 mg/kg of baclofen significantly reduced shortening responding. For both INT and D groups, 1.8 mg/kg baclofen reduced pellet and shortening responding for all schedules.
Buda-Levin et al., 2005 [39]	**Species:** Rat **Binge food:** Vegetable shortening, chow + shortening mixture. **Access:** Vegetable shortening for 2 h on Monday, Wednesday, and Friday (MWF), continuous access to chow + shortening mixture, or standard chow alone. **Frequency:** Daily (D) or Intermittent (INT) on non-consecutive days, 3 days a week. **Duration:** ~4 months **Calorie Restriction:** None (30 min post-injection)	Male Sprague Dawley rats were divided into 3 groups: Binge (B), Fat-Matched (FM), and Chow (C) groups. Animals were exposed to binge food in INT or D. The effect of the GABA-B agonist, baclofen (0, 0.3, 0.6, 1.0, 1.8 mg/kg IP), was assessed for the reduction of binge food intake.	With non-bingeing rats, baclofen generally had no effect on increased food intake. Baclofen (1.0, 1.8 mg/kg IP) reduced binge food intake regardless of access condition. Baclofen had no effect on FM or regular chow intake.
**Glutamatergic**			
**Study**	**Methods**	**Details**	**Major Findings**
Oliveira et al., 2021 [52]	**Species:** Mouse **Binge Food:** High-fat diet: (45% fat) **Access:** Ad libitum, assesed 2.5 h after caloric restriction. **Frequency:** Daily **Duration:** 15 weeks **Calorie Restriction:** One period of 14 h food deprivation at week 12 prior to drug treatment.	The metabotropic glutamate receptor 5 (mGluR5) is known to modulate central reward pathways. The effect of the mGluR5 negative allosteric modulator VU0409106 (3, 7.5, or 15 mg/kg) or FLX (10 mg.kg) in regulating feeding and obesity parameters in diet-induced obese C57BL/6 mice was assessed. Food intake, body weight, inflammatory/hormonal levels, and behavioral tests were measured/performed.	Reduction of feeding, body weight, and adipose tissue inflammation was seen in mice treated with high-fat diet after chronic treatment with VU0409106. VU0409106 (7.5 or 15 mg/kg) reduced binge-like eating.
Smith et al., 2015 [45]	**Species:** Rat **Binge Food:** Chocolate-flavored high-sucrose (50% kcal) AIN-76A-based pellets **Access:** 1 h **Frequency:** Daily **Duration:** >12 days **Calorie Restriction:** None	Male Wistar rats were trained on binge eating paradigm to operant respond (FR1) for binge food. Rats were administered an uncompetitive N-methyl-D-Aspartate antagonist, memantine (0, 1.25, 2.5, 5, and 10 mg/kg, IP), 30 min prior to the binge food or standard diet access. Another group of rats had cannula targeting the nucleus accumbens (NAcc) core or shell and received (0, 2.5, 10, and 20 µg/side) immediately before access.	Memantine (2.5–10 mg/kg IP) reduced binge intake; there was no effect on standard diet. Intra-accumbens injections of 10 and 20 µg reduced binge intake in animals targeting NAcc shell, but no effect was found with injections targeting the NAcc core.

#### 3.2.1. Noradrenergic/Adrenergic-Acting Compounds

Three compounds (prazosin, guanfacine, and nisoxetine) were used to target the central noradrenergic system in binge eating rodents [43,44,50]. Prazosin, an α-1 adrenergic receptor antagonist, increased the progressive ratio break point for binge food at all doses tested (0.5–2 mg/kg) in male rats [50]. Chronic guanfacine, an α2A-adrenergic receptor agonist (0.5 mg/kg/daily), increased binge eating in female rats exposed to limited binge food access without intermittent calorie restriction [43]. The selective norepinephrine reuptake inhibitor, nisoxetine (3 mg/kg), produced a feeding suppression in all groups, except the binge group with a history of intermittent calorie restriction [44].

#### 3.2.2. Dopaminergic-Acting Compounds

Three compounds (LDX, SCH23390, and raclopride) were used to target the central dopaminergic system [42,46,47,51]. LDX at 1.5 mg/kg reduced operant responding for binge food in female rats [51]. At 0.8 mg/kg, LDX normalized binge eating-induced delay discounting and binge eating-induced avoidance behaviors in female rats [46,47]. In male mice with deep brain stimulation (DBS)-induced binge eating, raclopride (3 mg/kg), a dopamine D2 receptor antagonist, effectively reduced DBS-induced binge eating [42]. In contrast, SCH23390 (0.75 mg/kg), a dopamine D1 receptor antagonist, was not effective in reducing DBS-induced binge eating [42].

#### 3.2.3. Serotonergic-Acting Compounds

Four compounds (FLX, fenfluramine, lorcaserin, and WAY163909) were used to target the central serotonergic system [38,41,48,49,53]. A selective serotonin reuptake inhibitor (SSRI), FLX was administered a single peripheral dose (30 mg/kg) or chronic dose (150 mg/L in drinking water for 21 days) to assess binge eating in mice with a knock-in of tryptophan hydroxylase 2 (*Tph2*) [53]. Peripherally administered FLX at 3 mg/kg reduced binge eating in female rats with a history of caloric restriction [41], whereas FLX at 2 mg/kg reduced binge eating in female rats exposed to acute foot shock stress [38]. Peripherally administered serotonergic drugs, such as fenfluramine (3 mg/kg), and lorcaserin (1 mg/kg), were used to assess intra-accumbens DAMGO-induced binge eating [48]. In addition, a novel 5HT2c receptor agonist, WAY163909, was effective at reducing binge eating at 1 mg/kg and at 2 mg/kg in male rats [49].

#### 3.2.4. γ-Aminobutyric Acid (GABA)-Ergic Compounds

The GABAergic system was targeted in two studies using the GABA_B_ agonist, baclofen, on binge eating in male rats [39,40]. Baclofen at 1 mg/kg reduced lever responding for binge food [40], whereas at 1.8 mg/kg baclofen reduced binge food intake under all access conditions [39].

#### 3.2.5. Glutamatergic Compounds

The glutamatergic system was targeted with two compounds (memantine and VU0409106) [45,52]. VU0409106, a negative allosteric modulator of the metabotropic glutamate receptor 5 (mGluR5), reduced binge eating in mice at 7.5 or 15 mg/kg [52]. The uncompetitive N-methyl-D-Aspartate antagonist, memantine, (2.5, 5, or 10 mg/kg), administered peripherally and site-specifically into the nucleus accumbens shell (10 or 20 µg), reduced binge eating in male rats [45].

### 3.3. Opioidergic Compounds

As described in Table 2, there were a total of five studies [54,55,56,57,58] that targeted the opioid system. The opioid antagonists, naloxone (0.1–1 mg/kg) and naltrexone (1 or 3 mg/kg), showed reductions in binge eating in rats following peripheral administration [54,55,56]. The novel opioid antagonist, GSK1521498 (1 or 3 mg/kg), reduced binge eating in male and female rats [56]. Peripherally administered naloxone (10 or 30 mg/kg) or intra- paraventricular hypothalamic nucleus naloxone (10–100 nmol) reduced intra-accumbens DAMGO-induced binge eating [58]. Two studies demonstrated the binge promoting effects of butorphanol (8 mg/kg), an opioid agonist/antagonist, on binge eating in female rats [55,57].

### 3.4. Hormone Analogs

As described in Table 3, there were a total of 12 studies that targeted hormonal signaling pathways [59,60,61,62,63,64,65,66,67,68,69,70].

#### 3.4.1. Estrogenic Compounds

The targets of sex steroids were examined in seven studies [59,60,61,62,63,64,65]. Several studies examined the effects of estradiol (E2) on binge eating behavior in ovariectomized (OVX) female mice. Results were mixed with E2 either reducing or not influencing binge eating behaviors [59,62,63,64,65]. E2 treatment combined with progesterone reduced binge food intake compared with the vehicle or progesterone alone [64,65]. However, the estrogen metabolite, 2-hydroxy-estradiol (2OH-E2), increased binge eating behavior in male and female rats [60,61].

#### 3.4.2. Corticotropin-Releasing Factor (CRF) and Stress-Related Compounds

Compounds targeting stress-related hormonal and brain signaling were examined in two studies [68,69]. Specifically, CRF receptor antagonists, centrally administered D-Phe-CRF _(12–41)_ or peripherally administered R121919 (20 mg/kg), reduced binge eating in rats with a history of calorie restriction and stress exposure [68,69]. Neither exogenous corticosterone nor metyrapone influenced binge eating [69].

#### 3.4.3. Gut Peptide Analogs

Three studies used compounds that targeted glucagon-like peptide-1 (GLP-1) and ghrelin signaling [66,67,70]. Intra-accumbens administration of GLP-1 receptor agonist (exendin 4; ex4; 0.05 and 0.1 µg) and antagonist (exendin 9; ex9; 2.5 and 5 µg) reduced and increased, respectively, mu-opioid-stimulated binge eating [70]. Neither intra-accumbens ex4 nor ex9 alone influenced binge food intake [70]. Acute central ghrelin (1 or 2 µg) reduced binge food preference, whereas chronic central ghrelin (0.5 μg/h for 28 days) increased binge food intake in male rats [66]. Central ghrelin signaling was further delineated by using exogenous LEAP2, an antagonist for ghrelin receptor, and K-(D-1-Nal)-FwLL-NH2, a growth hormone secretagogue receptor (GHSR) inverse agonist, to reduce binge eating in male mice [67].

### 3.5. Additional Individual Compounds

As described in Table 4, there were a total of 17 studies that examined additional compounds individually that were insufficient in the number of studies to justify a single table or category [71,72,73,74,75,76,77,78,79,80,81,82,83,84,85,86,87].

**Table 2 biomolecules-13-00742-t002:** Opioids and related compounds. Studies using opioidergic compounds and related targets.

Opioidergic			
Study	Methods	Details	Major Findings
Blasio et al., 2014 [54]	**Species:** Rat **Binge Food:** Chocolate-flavored high-sucrose (50% kcal) AIN-76A-based pellets **Access:** 1 h **Frequency:** Daily **Duration:** >12 days **Calorie Restriction:** None	Male Wistar rats were administered naloxone (0.03–0.3 mg/kg, SQ) or microinfused (5 µg or 25 µg/side) into the nucleus accumbens (NAcc) shell or medial frontal cortex (mPFC) to assess FR1 or breakpoint for binge food. Gene expression of opioid-related genes (*POMC, PDyn and PEnk*) was assessed in NAcc or mPFC.	SQ naloxone at 0.3 mg/kg reduced intake and breakpoint for standard chow and binge food. SQ naloxone at 0.1 mg/kg reduced chow and binge food intake and standard chow breakpoint, but not binge food breakpoint. Intra-NAcc at 25 µg reduced intake and breakpoint for standard chow and binge food. At 5 µg, it also reduced binge food breakpoint, but not intake. Intra-PFC binge food intake/breakpoint was suppressed at 25 µg. No effect was noted on chow intake/breakpoint. Intra-PFC at 5 µg did not have an effect on intake or breakpoint. POMC was increased and PDyn was decreased in mPFC.
Giuliano et al., 2012 [56]	**Species:** Rat **Binge food:** Chocolate-flavored pellets **Access:** 1 h chow access followed by 2 h food deprivation, 10 min access to chow feeder, and 10 min access to another chow feeder. **Frequency:** 15 consecutive days (preference) **Duration:** >2 months **Calorie Restriction:** 18 g/day lab chow for two weeks prior to the food-seeking experiment.	Male and female Lister Hooded rats were exposed to binge eating and food-seeking paradigms. The effects of the novel mu-opioid receptor antagonists, GSK1521498 (0.1, 1, and 3 mg/kg; SC) or Naltrexone (NTX, 0.1, 1, and 3 mg/kg; SC), on food seeking behavior (measured in rats trained to respond for reward under a second-order schedule of reinforcement) and binge-like eating were assessed.	Both GSK1521498 (1 or 3 mg/kg) and NTK (1 or 3 mg/kg) reduced binge-like eating. GSK1521498 reduced the anticipatory chow hypophagia. GSK1521498 (1 or 3 mg/kg) reduced food seeking both before and after binge food ingestion. NTX reduced food seeking only after binge food ingestion.
Naleid et al., 2007 [58]	**Species:** Rat **Binge food:** Choice of fat diet (77% fat) or sucrose diet (69% carbohydrate) **Access:** 3 h **Frequency:** Daily (D) **Duration:** ~3 weeks per experiment following one-week post-surgery recovery time. Total ~2 months. **Calorie Restriction:** 80% of free-feeding intake for NTX no-choice experiment only.	Male Sprague Dawley rats were implanted bilaterally with guide cannula in the PVN. The effects of the mu-opioid agonist DAMGO (0, 0.025, 0.25, and 2.5 nm, injected unilaterally to stimulate feeding) and the general opioid antagonist NTX (0, 10, 30, and 100 nmol, injected bilaterally) to inhibit feeding in consumption of fat and sucrose diets were assessed. The effect of peripheral subcutaneous injections of NTX (SQ; 0, 0.03, 0.1, and 0.3 mg/kg) was also assessed.	DAMGO increased intake of fat in the fat-preferring group but had no effect on intake of either diet in the sucrose-preferring group. NTX dose-dependently inhibited fat intake in both groups. Intra-PVN NTX did not inhibit sucrose intake when presented with a fat choice. NTX SQ (10 and 30 mg/kg) and intra-PVN (10, 30 and 100 nmol) inhibited intake of chow in 24 h food-deprived animals.
Boggiano et al., 2005 [55]	**Species:** Rat **Binge food:** Oreo cookies **Access:** Binge eating was produced in the R + S group through 3 cycles of restriction-refeeding, each of which lasted 12 days. Refeeding consisted of presentation of cookies for a short period, followed by ad lib chow. **Frequency:** 12-day cycle: 5 days ad lib chow or restriction, 2 days ad lib chow and cookies, 4 days ad lib chow, 1 day stress/no stress. **Duration:** 3–17 cycles of 12 days of restriction/refeeding followed by a test day on day 13.	Female Sprague Dawley rats were exposed to the binge paradigm with or without stress or restriction. The groups were as follows: Restriction and Stress (R + S), Restriction but No Stress (R + NS), Stress but No history of Restriction (NR + S), and a group with No History of Restriction or Stress (NR + NS) The effects of naloxone (1 mg/kg) and butorphanol (8 mg/kg) on binge eating were assessed.	Butorphanol (8 mg/kg) had exaggerated responses in the R + S rats. Butorphanol did not increase chow intake, but enhanced binge food intake. Naloxone (1 mg/kg) suppressed binge food in the R + S group to control levels.
Hagan & Moss, 1991 [57]	**Species:** Rat **Binge Food:** Standard chow **Access:** 1–4 h **Frequency:** Daily **Duration:** >1 year **Calorie Restriction:** Limited access to 4 h per day for first deprivation (DEP) episode, 1-to-3 h for 2nd and 3rd episodes.	Female Sprague Dawley rats were deprived for 3 episodes down to 75% normal body weight (1st episode) or 80% normal body weight (2nd and 3rd episodes) by limiting access to standard chow to 4 h per day (1st episode) or 2–3 h per day (2nd and 3rd episodes), followed by recovery to normal weight. Control group had ad libitum access. Once recovered, food was removed 40–60 min before testing. Intake of chow was measured 1, 2, and 3 h after butorphanol (SQ; 8 mg/kg) or vehicle.	History of DEP was highly significant on the amount eaten. Butorphanol increased food intake in rats with a history of caloric restriction/refeeding, but not in control rats.

**Table 3 biomolecules-13-00742-t003:** Hormone analogs and related compounds. Studies that use estrogenic/androgenic, stress-related, or gut peptides compounds or their combination.

**Estrogenic**			
**Study**	**Methods**	**Details**	**Major Findings**
Anversa et al., 2020 [59]	**Species:** Mouse **Binge Food:** Reese’s peanut butter drops and Nestlé chocolate drops. **Access:** 2 h for days 5–6 of the 8-day cycle **Frequency:** 8-day cycle: 1–4 days of calorie restriction, 5–8 days ad libitum feeding. Day 8 was the stress. **Duration:** 16 days, 2 cycles **Calorie Restriction:** 68% of ad libitum standard chow **Stress:** 15 min exposure to sight and smell of binge food without the ability to consume it. (*Frustration Stress*)	Male and female C57BL/6J mice. A subset of female mice received ovariectomy (OVX), OVX + estrogen replacement, sham, or control (no surgery or stress).	Male and female mice showed an increase in binge food intake when exposed to the cycles with Restriction + Stress. Females only demonstrated increase in binge food intake in the ad libitum feeding (i.e., no calorie restriction) + stress. No differences between groups were noted in the OVX, OVX + estrogen, and sham female mice.
Brutman et al., 2019 [62]	**Species:** Rat **Binge Food:** High-fat diet (40% fat) **Access:** 2 h **Frequency:** Every 3rd day (HFD-3D) or every day (HFD-ED) **Duration:** 45 days **Calorie Restriction:** None	Female Long Evans rats were exposed to HFD 3D, HFD-ED, or chow. Following the binge paradigm, rats were training for operant responding for sucrose pellets. A subset of rats underwent ovariectomy (OVX), or sham OVX, with or without estrogen (E2) treatment.	HFD-3D and HFD-ED did not differ in binge food intake. HFD-3D did not alter the motivation for sucrose. Overall, OVX and Sham had higher mean binge food intakes, and E2 had the lowest mean binge food intakes. E2 made more correct operant responses for sucrose than OVX + E.
Di Bonaventura et al., 2017 [63]	**Species:** Rat **Binge Food:** Sweetened hazelnut cocoa spread (52%), ground standard rat chow (33%), and 15% water. **Access:** 2 h for days 5–6 of the 8-day cycle **Frequency:** 8-day cycle: 1–4 days of calorie restriction, 5–8 days ad libitum feeding. **Duration:** 24 days, day 25 test day **Calorie Restriction:** 66% of ad libitum standard chow **Stress:** 15 min exposure to sight and smell of binge food without the ability to consume it. (*Frustration Stress*)	Female Sprague Dawley rats were assessed by the vaginal cytology to determine ovarian phase. Additionally, rats underwent ovariectomy and received estrogen (E2) or oil (OIL) to determine the influence on binge eating. Brain region expression of phosphorylated extracellular signal-regulated kinases (pERK) were assessed.	Diestrus or proestrus cycle stage had greater binge intake in Stress + Restriction group. OVX + E conditions resulted in lower binge intake than OVX + OIL in the Stress + Restriction group. pERK immunopositive cells were increased in the OVX + OIL in the Stress + Restriction group in PVN, CeA, and BNST. Cells counts were lower in OVX + E2 animals.
Yu et al., 2011 [65]	**Species:** Rat **Binge food:** Vegetable shortening. **Access:** 1 h **Frequency:** Intermittent 3 non-consecutive days (INT) per week. **Duration:** 4 weeks **Calorie Restriction:** None	Female ovariectomized Sprague Dawley rats were assigned to 1/3 of 4-day cyclic hormone treatments: estradiol (E) benzoate, progesterone (P), both, or oil vehicle, followed by an INT binge feeding protocol. The individual roles of estradiol (E) and progesterone (P) in the control of food intake and body weight in ovariectomized rats were assessed.	There were no overall effects of hormone treatments on binge food intake during the 1 h access period. E and E + P reduced fat intake relative to P during the 1 h fat access period during first binge access periods, but this difference diminished over time.
Babbs et al., 2013 [60]	**Species:** Rat **Binge Food:** Vegetable shortening **Access:** 1 h in week 1, 40 min in week 2, and 20 min in weeks 3 and 4. **Frequency:** 3 times a week on non-consecutives days **Duration:** 4 weeks **Calorie Restriction:** None	Female Sprague Dawley rats were ovariectomized (OVX) and received 2-hydroxyestradiol (2-OHE2) or vehicle via osmotic pump. Binge eating behavior was also compared with sham OVX (Intact). Binge eating male Sprague Dawley rats were implanted with microdialysis cannula targeting the prefrontal cortex (PFC) to measure dopamine (DA) efflux-administered acute vehicle or 2-OHE2 (3 μg/kg; IP).	Binge intake was increased in 2OHE2 and OVX compared with intact. 2OHE2-treated binge rats had lower standard chow intake and higher binge food to standard chow ratio. Binge food increase in DA efflux was abolished in binge rats receiving 2OHE2.
Babbs et al., 2011 [61]	**Species:** Rat **Binge food:** Vegetable shortening **Access:** 1 h **Frequency:** Daily (D) or Intermittent (INT) on non-consecutive days, 3 days a week. **Duration:** >5 weeks **Calorie Restriction:** None	Two cohorts (1 male and 1 female) of Sprague Dawley rats were exposed to a daily (D) or intermittent (INT) limited-access, binge eating paradigm for 5 weeks. Assessment of the estrogen system on binge eating behavior of 2OHE2 (1, 3, and 10 ug/kg; IP), vehicle, or 2-methoxyestradiol (2ME2; 3.0 µg/kg; IP) immediately prior to fat access following the 5-week binge-induction period.	Males consumed significantly less fat/kg of body weight than did females after 5 weeks. Binge intake was significantly stimulated by 2OHE2 at 3 µg/kg only in the INT group. 2ME2 had no effect on binge food intake. There was no effect of 2OHE2 and 2ME2 on binge food intake in D access groups.
Yu et al., 2008 [64]	**Species:** Rat **Binge food:** Vegetable shortening **Access:** 1 h **Frequency:** Daily (D; low restriction) or Intermittent (3 non-consecutive days; INT; high restriction) per week. **Duration:** 6 weeks **Calorie Restriction:** None	Ovariectomized female Sprague Dawley rats were exposed to a binge feeding paradigm of either chow only, INT, or D. The effect of ovarian hormones E (estradiol benzoate and P (progesterone), 2 ug/100 µL and 500 µg/100 µL sesame oil or oil vehicle, respectively, injected every 4 days) on binge feeding behavior was assessed.	INT groups showed binge-like intake over time. E + P-treated rats consumed significantly less calories compared with vehicle-treated rats. E + P treatment produced a sustained reduction in binge food intake in D group, whereas INT had a variable (i.e., cyclical) suppression over the binge food access periods.
**Corticotropin-releasing factor (CRF)/Stress-Related Studies**			
**Study**	**Methods**	**Details**	**Major Findings**
Di Bonaventura et al., 2017 [69]	**Species:** Rat **Binge Food:** Sweetened hazelnut cocoa spread (52%), ground standard rat chow (33%), and 15% water. **Access:** 2 h for days 5–6 of the 8-day cycle **Frequency:** 8-day cycle: 1–4 days of calorie restriction, 5–8 days ad libitum feeding. **Duration:** 24 days, day 25 test day **Calorie Restriction:** 66% of ad libitum standard chow **Stress:** 15 min exposure to sight and smell of binge food without the ability to consume it. (*Frustration Stress*)	Female Sprague Dawley rats were exposed to the binge eating paradigm with CRF1 receptor antagonist R121919 (IP; 0,10. 20 mg/kg) or metyrapone (IP; 0, 50, 100 mg/kg). Non-stressed rats were treated with corticosterone (IP; 2.5 mg-15 mg/kg). In another set of rats, *crhr1* mRNA was assessed in the BNST, CeA, BLA, and paraventricular nucleus. Lastly, binge eating was assesed following CeA or BLA injections of the CRF receptor antagonist D-Phe-CRF _(12–41)_ (300 ng/side).	R121919 (20 mg/kg) reduced binge intake in Restriction + Stress group, whereas metyrapone did not influence binge eating. Corticosterone did not result in a binge eating phenotype. Dorsal BNST and CeA had increased *crhr1* in the Restriction + Stress group. D-Phe-CRF _(12–41)_ in the CeA blocked the 15 min binge eating.
Di Bonaventura et al., 2014 [68]	**Species:** Rat **Binge Food:** Sweetened hazelnut cocoa spread (52%), ground standard rat chow (33%), and 15% water. **Access:** 2 h for days 5–6 of the 8-day cycle **Frequency:** 8-day cycle: 1–4 days of calorie restriction, 5–8 days ad libitum feeding. **Duration:** 24 days, day 25 test day **Calorie Restriction:** 66% of ad libitum standard chow **Stress:** 15 min exposure to sight and smell of binge food without the ability to consume it. (*Frustration Stress*)	Female Sprague Dawley rats were exposed to a binge eating paradigm with or without stress or restriction. The systemic effects of the CRF-1 receptor antagonist R121919 (10 or 20 mg/kg, SQ) were assessed. Intraventricular injections (ICV) or intra-BNST injection of nonselective CRF receptor antagonist D-Phe-CRF _(12–41)_ (ICV dose 100–1000 ng; BNST dose; 10–50 ng/side) were also determined. Regional C-Fos immunoreactivity (IR) was assessed.	Rats exposed to the Restriction + Stress paradigm had greater binge food intake. Systemic R121919 (20 mg/kg) reduced binge food intake in Restriction + Stress group. Increased c-Fos-IR was observed in the dorsal and ventral BNST in the Restriction + Stress rats. Intra-BNST injections of D-Phe (25 ng and 50 ng) reduced binge intake in 15 min in the Restriction + Stress group. At 2 h, only the 50 ng effectively reduced binge intake. ICV injections of D-Phe-CRF _(12–41)_ at 1000 ng effectively reduced binge intake at 15 min.
**Gut peptides**			
**Study**	**Methods**	**Details**	**Major Findings**
Pierce-Messick and Pratt, 2020 [70]	**Species:** Rat **Binge Food:** High-fat, high-sucrose diet **Access:** 2 h **Frequency:** Daily (D) **Duration:** ~6 weeks **Calorie Restriction:** None	Male Sprague Dawley rats were implanted with bilateral cannula above the nucleus accumbens core. The effects of nucleus accumbens injections of the GLP-1 receptor agonist exendin 4 (EX4) (0, 0.05, and 0.1 µg/side) of antagonist exendin 9 (EX9) (0, 2.5, and 5.0 µg/side) on binge food consumption with and without simultaneous mu-opioid receptor stimulation (via DAMGO; 0.25 ug/side) was assessed.	Intra-accumbens Ex4 or Ex9 alone did not influence binge food intake. Intra-accumbens DAMGO-induced binge eating was attenuated when combined with both doses of Ex4. Dynamics of intra-accumbens DAMGO-induced binge-like feeding was altered when combined with Ex9. Ex9 extended DAMGO-induced binge eating to increase binge intake.
Cornejo et al., 2019 [67]	**Species:** Mouse **Binge Food:** High palatable pellets (21.1% fat, 22.5% carbohydrates) **Access:** 2 h **Frequency:** Daily **Duration:** 4 consecutive days **Calorie Restriction:** None	Male C57BL/6J mice were investigated for growth hormone secretagogue receptor (GHSR) activity. Ghrelin (300 pmol; ICV) and LEAP2 (liver-expressed antimicrobial peptide 2; 600 pmol/g SQ; 600 pmol ICV), [D-Lys3]-GHRP-6 (2 nmol ICV), JMV2959, (3 nmol ICV), and K-(D-1-Nal)-FwLL-NH2 (1 nmol; ICV) were used to investigated GHSR in the periphery and brain.	Plasma levels of ghrelin and LEAP2 were not altered with the binge eating protocol. Central, but not peripheral, injections of the LEAP2 analog reduced the binge intake. Similarly, central administration of K-(D-1-Nal)-FwLL-NH2, GHSR inverse agonist also reduce binge intake. Central ghrelin, JMV2959, and [D-lys3]- GHRP-6 did not influence binge eating.
Bake et al., 2017 [66]	**Species:** Rat and Mouse **Binge Food:** High-fat diet (60% Fat) **Access:** 2 h **Frequency:** Daily **Duration:** >2 weeks **Calorie Restriction:** None	Male Sprague Dawley rats were implanted with intracerebroventricular (ICV) cannulae or intra-VTA cannulae and administered acute ghrelin (1 or 2 µg/mL) prior to the binge. In a subset of rats, chronic ICV ghrelin (0.5 μg/h for 28 days) with 10 days standard chow/18 days on binge paradigm was evaluated. Another set of ghrelin receptor knockout (GHS-R KO) mice were examined for their binge phenotype.	In rats, acute ghrelin ICV or intra-VTA reduced binge food intake but increased standard chow intake during the 2 h binge access period. Chronic ICV ghrelin resulted in increased binge food intake and weight gain for the 18-day binge paradigm. Standard chow intake was not increased. GHS-R KO mice had lower body weight and reduced fat mass. Binge eating paradigm resulted in higher body weight, regardless of mouse strain.

**Table 4 biomolecules-13-00742-t004:** Additional compounds. Studies using PACAP, orexin, and other novel targeting compounds.

**Pituitary Adenylate Cyclase-Activating Polypeptide (PACAP)**			
**Study**	**Methods**	**Details**	**Major Findings**
Le et al., 2021 [80]	**Species:** Mouse **Binge Food:** High-fat diet **Access:** 1 h **Frequency:** Daily **Duration:** 5 days **Calorie Restriction:** None	Male and female C57BL/6/J (WT) tyrosine hydroxylase (TH-CRE) and pituitary adenylate cyclase-activating polypeptide (PACAP:CRE) mice (on a C57BL6/J background) were used. Experiments were designed to examine the ventromedial nucleus (VMN) PACAP neurons projects to the ventral tegmental area (VTA) and binge eating. Female WT mice were ovariectomized (OVX) and received vehicle or SQ estrogen daily. Immediately before binge access, WT mice were injected with PACAP1–38 (30 pmol, 0.2 μL) or 0.9% saline vehicle (0.2 μL) into the VTA. Electrophysiological recordings from TH-CRE and PACAP-CRE were taken.	Intra-VTA PACAP suppressed binge intake in WT males but not in female intact or OVX mice. Electrophysiological of PACAP neurons VMN projecting to VTA DA neurons were attenuated by the KATP channel blocker tolbutamide and PAC1 receptor antagonist PACAP6–38, suggesting that PACAP activation is associated with a hyperpolarization and decrease in firing of DA VTA neurons.
Hurley et al., 2016 [77]	**Species:** Rat **Binge Food:** Western diet (41% Fat) **Access:** 15 min (meal 2; M2) **Frequency:** Daily **Duration:** 2 weeks **Calorie Restriction:** Daily 2 h standard chow access (meal 1; M1), 30 min before M2.	Male Sprague Dawley rats were implanted intra-ventromedial hypothalamus (VMN) and intra-nucleus accumbens (NAc) cannula. Binge eating paradigm consisted of M2 as binge food or standard chow as control. Pituitary Adenylate Cyclase-Activating Polypeptide (PACAP; 50 pmol/side) was injected in the either intra-VMN or intra-NAc to investigate the regional PACAP influence on binge-like eating.	Intra-VMN, PACAP, and AMPA before M1 reduced standard chow (M1) intake. Intra-NAc, PACAP, or baclofen + muscimol (106.8 ng/5.7 ng/side) reduced binge food intake (M2) injected either before or after M1.
**Orexin**			
**Study**	**Methods**	**Details**	**Major Findings**
Schuler and Koch, 2022 [87]	**Species:** Rat **Binge Food:** Vegetable fat **Access:** 1 h **Frequency:** Daily (non-binge) or Every other day (binge) **Duration:** 6 weeks **Calorie Restriction:** None	Male Lister Hooded rats were trained and screened for impulsivity on a five-choice serial reaction time task. After exposure to the binge eating paradigm, rats were bilaterally implanted with guide cannula target the NAC shell. Rats received microinfusions of either vehicle, orexin 1 receptor anatagonist (SB-338467; 500 ng/0.3 μL), or cocaine-and amphetamine-regulated transcript antibodies (CARTaB; 1:500) immediately before acess to the binge food. Microinfusions were 2–3 days apart.	SB-338467 in NAC shell reduced binge food intake in the non-binge exposure compared with vehicle. Based on impulsivity ranking, SB 338467 only reduced binge food intake in the non-binge paradigm in rats with low impulsivity. CARTaB in the NAC shell did not alter binge food intake.
Valdivia et al., 2015 [85]	**Species:** Mouse **Binge Food:** High-fat diet (HFD: 22.5% carbohydrate, 22.8% proteins, 21.1% fat, 23.0% fibers, 5.6% minerals, and 5.0% humidity). **Access:** 2 h **Frequency:** Daily **Duration**: 4 days **Calorie Restriction:** None	C57BL/6J mice (WT; wild type) were exposed to the 1 or 4 day of binge food, ad libitum binge food, or standard chow (no binge food). Regional immunoreactivity (IR) for tyrosine hydroxylase (TH), orexin, or c-Fos was determined. In separate experiments the influence of orexin, by injection of orexin 1 selective antagonist SB-334867 (5 mg/Kg, IP), and ghrelin, by use of growth hormone secretagogue receptor (GHSR)-null mice, were determined.	Mice exposed to intermittent binge food access escalated intake by day 4. TH-IR was increased in subdivision of the VTA; the interfascicular nucleus (IF) and parabrachial pigmented (PBP) area on day 4 of the binge eating rats. c-Fos-IR was increased in nucleus accumbens: medial shell, lateral shell core, and core on day 4 of the binge eating rats. c-Fos-IR and orexin-IR were increased in the lateral hypothalamus. SB-334867 treatment only altered binge intake on day 1 and did not prevent the escalation in intake over 4 days. GHSR-null mice had reduced binge intake and reduced TH-IR in the subregions of the VTA and reduced c-Fos-IR in the subregions of nucleus accumbens.
Piccoli et al., 2012 [83]	**Species:** Rat **Binge food:** Paste mixture of Nutella (52%), food pellets (33%), and water (15%). **Access:** 2 h for days 5–6 of the 8-day cycle **Frequency:** Three 8-day cycles: 1–4 days of calorie restriction, 5–8 days ad libitum feeding. **Duration:** 24 days, day 25 test day. **Calorie Restriction:** 4 days per week at 66% restriction of ad libitum chow. **Stress:** 15 min exposure to sight and smell of binge food without the ability to consume it. (*Frustration Stress)*	Female Sprague Dawley rats were exposed to a binge eating paradigm of restriction (R) and stress (S). Effects of GSK1059865 (a selective OX1R antagonist) (10 and 30 mg/kg; IP), JNJ-10397049 (a selective OX2R antagonist) (1, 3 mg/kg; IP), and SB-649868 (a dual OX1R/OX2R receptor antagonist) (3, 10 mg/kg; gavage) on binge eating were assessed.	SB-649868 and GSK1059865 selectively reduced binge-like eating in R + S group. JNJ-10397049 was not effective in reducing binge eating behavior.
**Sigma (σ) receptors**			
**Study**	**Methods**	**Details**	**Major Findings**
Cifani et al., 2020 [72]	**Species:** Rat **Binge Food:** Sweetened hazelnut cocoa spread (52%), ground standard rat chow (33%), and 15% water. **Access:** 2 h for days 5–6 of the 8-day cycle **Frequency:** 8-day cycle: 1–4 days of calorie restriction, 5–8 days ad libitum feeding. **Duration:** 24 days, day 25 test day **Calorie Restriction:** 66% of ad libitum standard chow **Stress:** 15 min exposure to sight and smell of binge food without the ability to consume it. (*Frustration Stress*)	Female Sprague Dawley rats were exposed to the binge paradigm or three control conditions: No calorie restriction, No Stress, or No calorie and No Stress. A potent novel sigma1 (σ1), receptor antagonist,1,3-dioxane derivative, 9-Benzyl-3-phenyl-1,5-dioxa-9-azaspiro[5.5]undecane, was evaluated in binge eating rats.	Novel σ1 receptor antagonist (3 mg/kg and 7 mg/kg; IP) suppressed binge food intake in rats expose to the binge eating paradigm. No effects were observed on performing the open field and forced swimming tests with the novel compound.
Del Bello et al., 2019 [74]	**Species:** Rat **Binge Food:** Sweetened hazelnut cocoa spread (52%), ground standard rat chow (33%), and 15% water. **Access:** 2 h for days 5–6 of the 8-day cycle **Frequency:** 8-day cycle: 1–4 days of calorie restriction, 5–8 days ad libitum feeding. **Duration:** 24 days, day 25 test day **Calorie Restriction:** 66% of ad libitum standard chow **Stress:** 15 min exposure to sight and smell of binge food without the ability to consume it. (*Frustration Stress*)	Female Sprague Dawley rats were dosed with a novel σ1 receptor antagonist (1, 3, and 7 mg/Kg). Compound was the racemic spipethiane analogue (2-(1-Benzylpiperidin-4-yl)thiochroman-4-one) (±)-1.	Binge eating was observed in the rats with Restrict + Stress conditions. Binge eating was reduced with the 3 mg/kg and 7 mg/kg dose of the σ1 receptor antagonist. 26.4% (38/144) of the rats were in estrous and did not demonstrate binge eating behavior, which were excluded from the statistical analyses.
**Other targeting compounds**			
**Study**	**Methods**	**Details**	**Major Findings**
Kania et al., 2020 [79]	**Species:** Rat **Binge Food:** Sweetened hazelnut cocoa spread (52%), ground standard rat chow (33%), and 15% water. **Access:** 2 h for days 5–6 of the 8-day cycle **Frequency:** 8-day cycle: 1–4 days of calorie restriction, 5–8 days ad libitum feeding. **Duration:** 36 days; test day (day 37). **Calorie Restriction:** 66% of ad libitum standard chow **Stress:** 15 min exposure to sight and smell of binge food without the ability to consume it. (*Frustration Stress*)	Female Sprague Dawley rats were exposed to the binge paradigm or control conditions (No calorie restriction on days 1–4 of the 8-day cycle and No Stress). Relaxin-family peptide-3 receptor (RXFP3) antagonist R3(B1-22)R (1 ug/0.5 uL in ACSF, 0.5 uL bilateral) or vehiclewere injected into the hypothalamic paraventricular nucleus (PVN; magnocellular region) 15 min before access to the binge food on test day.	Bilateral intra-PVN injections RXFP 3 antagonist (15 min pre-treating) reduced the amount of binge food intake during 2 h access period on test day (30 min and 120 min time points). No effects of intra-PVN RXP3 antagonist on consumption of binge food by control rats.
Romano et al., 2020 [84]	**Species:** Rat **Binge Food:** Sweetened hazelnut cocoa spread (52%), ground standard rat chow (33%), and 15% water. **Access:** 2 h for days 5–6 of the 8-day cycle **Frequency:** 8-day cycle: 1–4 days of calorie restriction, 5–8 days ad libitum feeding. **Duration:** 24 days, day 25 test day **Calorie Restriction:** 66% of ad libitum standard chow **Stress:** 15 min exposure to sight and smell of binge food without the ability to consume it. (*Frustration Stress*)	Female Sprague Dawley rats were exposed to the binge paradigm or three control conditions: No calorie restriction, No Stress, or No calorie and No Stress. Oleoylethanolamide (OEA) was dose-dependently administered (2.5, 5, and 10 mg/kg; IP) to suppress binge eating. OEA effects on binge eating rats were measured by neural activation, c-Fos activation, oxytocin receptor expression, and monoamine turnover. OEA effect on oxytocin on corticotropin-releasing factor (CRF) and oxytocin mRNA in hypothalamic and extrahypothalamic regions were also examined.	OEA (5 mg/kg) reduced binge eating at 15 min in rats exposed to the Restriction + Stress paradigm. OEA (10 mg/kg) reduced binge eating over the whole 2 h binge session in Restriction + Stress group. C-Fos immunostaining was reduced in nucleus accumbens (Acb), cauduate putamen (CPu), substania nigra (SN), and amygdala (AMY) by OEA in Restriction + Stress rats. Increased c-Fos activation of was observed in the ventral tegmental (VTA) and hypothalamic paraventricular nucleus (PVN) by OEA in Restriction + Stress rats. CRF was decrease in the central amygdala, while an increase in oxytocin mRNA levels was induced in the PVN by OEA in Restriction + Stress rats. Oxytocin receptor density were normalized in the Acb and CPu by OEA in Restriction + Stress rats.
Feltmann et al., 2018 [75]	**Species:** Rat **Binge Food**: Chocolate-flavored sucrose pellets **Access:** 10 min **Frequency:** daily **Duration:** 48 days **Calorie Restriction:** 2 h prior to binge food.	Male Lister Hooded rats were exposed to a binge paradigm of negative anticipatory contrast (10 min chow/10 min binge food). After stable intake, the dose-dependent effects of a novel monoamine stabilizer OSU6162 (SQ; 5, 10, 15, and 30 mg/kg) were assessed.	OSU 6162 at 15 mg/kg and 30 mg/kg reduced binge intake. Standard chow was not altered.
Ferragud et al., 2017 [76]	**Species:** Rat **Binge Food:** Chocolate-flavored, high-sucrose (50% kcal) AIN-76A-based diet (5TUL: 66.7% carbohydrate, 12% fat) **Access:** 1 h **Frequency**: Daily **Duration:** >15 days **Calorie Restriction:** None	Male Wistar rats were administered a highly selective Trace Amine-Associated Receptor 1 (TAAR1) full-agonist RO5256390 (0, 1, 3, 10 mg/kg, IP) 30 min prior to binge food access.	R05256390 reduced binge-like eating at 3 and 10 mg/kg from vehicle treatment. At 10 mg/kg RO5256390 binge food intake was similar to standard chow but did not alter food-restricted standard chow intake.
Hurley et al., 2016 [78]	**Species:** Rat **Binge Food:** Western diet (41% Fat) **Access:** 30 min **Frequency:** Daily **Duration:** 2 weeks **Calorie Restriction:** None	Male Sprague Dawley rats were dosed with N-acetylcysteine (NAC) or saline by systemic (90 mg/kg IP) or intraventricular (10 μg ICV) administration. Binge food intake and saccharin (0.15%) conditioned taste aversion (CTA) were assessed.	Systemic or ICV administration by NAC reduced binge food intake. NAC did not result in a reduction of standard chow or produce a CTA.
Bharne et al., 2015 [71]	**Species:** Rat **Binge Food:** Hhigh-fat, sweet palatable diet (HFSPD; 40% fat as vegetable shortening, 40% protein, and 20% carbohydrate with sweetened milk) **Access:** 2 h **Frequency:** Every other day **Duration:** 4 weeks **Calorie Restriction:** None	Male Wistar rats were implanted with lateral ventricle ICV cannula and the effects of cocaine- and amphetamine-regulated transcript peptide (CART; 1, 1.5 and 2 µg/rat), CART antibody (1:500 and 1:250 dilution, 5 µL/rat), and the CB1 inverse agonist rimonabant (1, 3, and 10 mg/kg; IP) before binge food access. Morphometric analysis of CART immunoreactivity in feeding- and reward-related brain areas was investigated.	CART (1.5 and 2 µg) and rimonabant (3 and 10 mg/kg) reduced binge intake. CART (0.5 and 1 µg) and rimonabant (0.3 and 1 mg/kg) reduced chow intake. CART antibody (1:250) increased chow intake. CART cells and fibers in LH and ARC were increased 1 h mid-binge and decreased 22 h post-binge. PVN had decreased CART cells 22 h post-binge. CART fibers in nucleus accumbens shell (Acb Sh) and paraventricular nucleus (PVT) of thalamus were increased 1 h mid-binge and decreased 22 h post-binge. Rimonabant 1 h mid-binge increased cells and fibers in the ARC, increased fibers in PVT, and decreased fibers in Acb Sh. Immunoreactivity for CART + synaptophysin was increased in LH and CA1 hippocampus in binge eating rats.
Freund et al., 2015 [86]	**Species:** Rat **Binge Food:** All-vegetable shortening. **Access:** 1 h **Frequency:** Intermittent on non-consecutive days, 3 days a week. **Duration:** 5 weeks **Calorie Restriction:** None	Male and female Sprague Dawley rats received the tricyclic antidepressant clomipramine (CMI, 15 mg/kg; IP; BID) or vehicle during postnatal day (P) 9–16. Spontaneous alternation task, elevated plus maze, and marble burying were performed at three developmental time points P25, P60, and P100. In a separate experiment, rats were observed for sucrose pellet consumption, which began on P60 and P100. In another experiment, rats were exposed to binge eating paradigm on P55. Rats were also classified as binge-prone based on the upper 25% of consumption of chow.	CMI-treated rats displayed increased anxiety behaviors with age, P100 rats displayed the most anxiety-like behaviors. CMI-treated rats consumed less sucrose pellets. CMI-treated rats increased binge food intake in the last week of the binge eating paradigm. 3 of the 8 CMI-treated males were binge-prone. No CMI-treated females were binge-prone.
Di Bonaventura et al., 2013 [82]	**Species:** Rat **Binge Food:** Sweetened hazelnut cocoa spread (52%), ground standard rat chow (33%), and 15% water. **Access:** 2 h for days 5–6 of the 8-day cycle **Frequency:** 8-day cycle: 1–4 days of calorie restriction, 5–8 days ad libitum feeding. **Duration:** 24 days, day 25 test day **Calorie Restriction**: 66% of ad libitum standard chow **Stress:** 15 min exposure to sight and smell of binge food without the ability to consume it. (*Frustration Stress)*	Female Sprague Dawley rats were exposed to a binge paradigm with or without stress or restriction. The central effects of nociception/orphanin FQ (N/OFQ) on binge eating were determined by intracerebroventricular (ICV) injection of N/OFQ (0.5-1 nmol/rat) or regionally mRNA by in situ hybridization of NOP receptor and ppN/OFQ.	ICV N/OFQ at 0.5 nmol/rat reduced 30 min binge intake in Restriction + Stress group. ICV N/OFQ at 1.0 nmol/rat increase binge intake in Restrict and no Stress group. ICV N/OFQ at 0.25 nmol/rat increased binge food intake in rats with repeated cycles of Restriction. Repeated bouts of restriction increased NOP receptor mRNA in the VMH. Increased ppN/OFQ mRNA was observed in VTA and BNST in rats exposed to Restriction
Di Bonaventura et al., 2019 [88]	**Species:** Rat **Binge Food:** Sweetened hazelnut cocoa spread (52%), ground standard rat chow (33%), and 15% water. **Access:** 2 h for days 5–6 of the 8-day cycle **Frequency:** 8-day cycle: 1–4 days of calorie restriction, 5–8 days ad libitum feeding. **Duration:** 24 days, day 25 test day **Calorie Restriction**: 66% of ad libitum standard chow **Stress:** 15 min exposure to sight and smell of binge food without the ability to consume it. (*Frustration Stress)*	Female Sprague Dawley rats received a single dose of a A_2A_ adenosine receptor antagonist ANR 94 (1 mg/kg), agonist VT 7 (0.1 mg/kg, IP), combination (pretreatment ANR 94 followed by VT 7 15 min later, IP), or vehicle before binge food access. Binge eating intake effects werestudied on A_2A_ adenosine receptor gene expression and DNA methylation in the amygdaloid complex.	VT7 alone reduced binge eating behavior. Amygdaloid A2A adenosine receptor gene expression was increased from the vehicle after VT 7 alone. ANR 94 + VT7 significantly reduced A2A adenosine receptor gene expression from VT 7 alone. ANR 94 alone and ANR 94 + VT7 increase % DNA methylation at adenosine receptor gene promoter compared with vehicle. %DNA methylation with ANR 94 + VT7 was less than VT7 alone.
Di Bonaventura et al., 2012 [81]	**Species:** Rat **Binge food:** Paste mixture of Nutella (52%), food pellets (33%), and water (15%). **Access:** 2 h for days 5–6 of the 8-day cycle **Frequency** Three 8-day cycles: 1–4 days of calorie restriction, 5–8 days ad libitum feeding. **Duration:** 24 days, day 25 test day. **Calorie Restriction:** 4 days per week at 66% restriction of ad libitum chow. **Stress:** 15 min exposure to sight and smell of binge food without the ability to consume it. (*Frustration Stress)*	Female rats were exposed to a restricting(R)/refeeding binge eating paradigm with or without stress (S). The effect of the A_2_A adenosine receptor (AR) agonists, CGS 21680 and VT 7 (0.1 and 0.05 mg/kg IP), on intake of highly palatable food (HPF) on sated rats, and low-palatability food (LPF) intake in food-deprived rats was assessed.	Rats in the R + S group had a higher intake of binge food than the NR + NS controls. CGS 21680 slightly reduced locomotor activity at 0.1 mg/kg; VT 7 did not modify locomotor activity. CGS 21860 (0.1 mg/kg) and VT7 (0.1 mg/kg) reduced binge food intake in both R + S and NR + NS groups. Both CGS 21860 (0.1 mg/kg) and VT 7 (0.1 mg/kg) reduced the standard LPF intake following a 24 h food deprivation.
Cifani et al., 2010 [73]	**Species:** Rat **Binge food:** Paste mixture of Nutella (52%), food pellets (33%), and water (15%). **Access:** 2 h for days 5–6 of the 8-day cycle **Frequency:** 8-day cycle: 1–4 days of calorie restriction, 5–8 days ad libitum feeding. **Duration:** 24 days, day 25 test day. **Calorie Restriction:** 4 days per week at 66% restriction of ad libitum chow. **Stress:** 15 min exposure to sight and smell of binge food without the ability to consume it. (*Frustration Stress)*	Female Sprague Dawley rats were exposed to a model of binge eating by three 8-day cycles of food restriction/refeeding (R) and stress (S). The effect of *Rhodiola rosea* dry extract (3% rosavin, 3.12% salidroside) in this model of binge eating was assessed. Additionally, the bioactive constituents of rosavin and salidroside were assessed.	Rats exposed to R + S conditions exhibited binge eating behavior in the first 15–60 min Reductions of binge eating were noted with *R. rosea* extract (10 mg/kg or 20 mg/kg) at 15- and 60-min during binge access. *R. rosea* extract at 20 mg/kg also reduced binge eating at 30 min and reduced stress-induced serum corticosterone levels. Bioactive components of *R. rosea*, salidroside (312–936 µg/kg), dose-dependently reduced binge eating.

#### 3.5.1. Pituitary Adenylate Cyclase-Activating Polypeptide (PACAP)

Two studies investigated site-specific injections of PACAP or PACAP_1–38_ [77,80]. Intra-accumbens PACAP (50 pmol/side) reduced binge eating, whereas ventromedial hypothalamic injections did not influence binge food intake in male rats [77]. Intra-VTA PACAP_1–38_ (30 pmol/side) reduced binge eating in male mice, but not in female or OVX female mice [80].

#### 3.5.2. Orexin Receptor-Targeting Compounds

Two studies investigated peripheral injections of selective orexin receptor-1 (OX_1_) or orexin receptor-2 (OX_2_) or combined (OX_1_/OX_2_) antagonists [83,85]. A selective OX_1_ antagonist, GSK1059865, (10 or 30 mg/kg), and a dual OX_1_/OX_2_ receptor antagonist, SB-649868 (3 mg/kg), reduced binge eating in rats with stress exposure [83]. A selective OX_2_ antagonist, JNJ-10397049 (1 or 3 mg/kg), was not effective in reducing binge food intake [83]. In contrast, the selective OX_1_ antagonist, SB-334867 (5 mg/kg), only reduced intake for the first binge eating episode, but was not effective at preventing the escalation in binge food intake over time in mice [85]. In one study examining the centrally administered OX_1_ antagonist, bilateral microinfusions of SB-334867 (500 ng/0.3 μL per side) into the nucleus accumbens shell were ineffective at reducing binge eating in male rats with high or low impulsivity [87].

#### 3.5.3. Sigma (σ) Receptors

Two studies examined novel sigma receptor antagonists in female rats with a history of calorie restriction and stress exposure [72,74]. Both the racemic spipethiane analogue (2-(1-Benzylpiperidin-4-yl)thiochroman-4-one) (±)-1 (3 or 7 mg/kg) and 1,3-dioxane derivative, 9-Benzyl-3-phenyl-1,5-dioxa-9-azaspiro[5.5]undecane (3 or 7 mg/kg) were effective in reducing binge eating behaviors [72,74].

#### 3.5.4. Other Targeting Compounds

Site-specific injections into the PVN of the relaxin-family peptide-3 receptor (RXFP3) antagonist, R3(B1-22)R (1 µg/side), reduced binge eating in female rats [79]. Peripheral administration of oleoylethanolamide (OEA; 10 mg/kg) reduced binge eating in female rats with stress exposure [84]. Peripheral injections of a novel monoamine stabilizer OSU6162 (15 or 30 mg/kg) reduced binge eating in male rats [75].

A selective trace amine-associated receptor 1 (TAAR1) full-agonist RO5256390 (3 or 10 mg/kg) administered peripherally reduced binge food intake in male rats [76]. Peripheral (90 mg/kg) or central administration of N-acetylcysteine (10 µg ICV) reduced binge eating without producing a conditioned taste aversion to the binge food or sodium saccharin solution [78]. Central injections of the cocaine- and amphetamine-regulated transcript peptide (CART; 1.5 or 2 µg; ICV) reduced binge eating, whereas the central injections of the CART antibody (1:500 or 1:250) did not influence binge food intake [71]. In the same study, peripheral injections of the inverse agonist of the cannabinoid receptor 1 (CB1), rimonabant (3 or 10 mg/kg), reduced binge intake [71].

Peripheral administration of the tricyclic antidepressant clomipramine twice daily (15 mg/kg) prior to weaning resulted in adult binge eating phenotype in male and female rats [86]. Examination of nociception/orphanin FQ (N/OFQ) on binge eating indicated a differential dose-dependent effect [82]. Central injection of N/OFQ peptide (0.5 nmol; ICV) decreased binge intake at 30 min in rats with a history of calorie restriction and stress, but higher doses (1.0 nmol ICV) increased binge eating in rats with a history of calorie restriction without stress [82].

Peripheral administration of two of the A_2A_ adenosine receptor agonists, CGS 21680 (0.1 mg/kg) and VT 7 (0.1 mg/kg), reduced binge food and non-binge food intake [81]. VT7 (0.1 mg/kg) also increased A2A receptor gene expression in the amygdaloid complex, whereas the A_2A_ antagonist ANR 94 (1 mg/kg) increased %DNA methylation of the A_2A_ gene promoter [88]. ANR94 combined with VT 7 had higher %DNA methylation than VT 7 alone [88]. Peripheral administration of extracts of the botanical *Rhodiola rosea* (10 or 20 mg/kg) reduced binge eating in female rats exposed to a history of calorie restriction and stress [73]. Additional experiments indicated salidroside (312–936 µg/kg), a bioactive component of *R. rosea*, reduced binge eating [73].

### 3.6. Combinational or Direct Comparisons of Multiple Compounds Acting on Different Systems

As described in Table 5, there were 16 studies that were combinational or were designed to be direct comparisons between compounds [89,90,91,92,93,94,95,96,97,98,99,100,101,102,103,104].

#### 3.6.1. Fluoxetine (FLX) Compared with Other Compounds

Six studies [89,90,91,92,94,98] compared FLX with other compounds. In all studies, FLX was administered peripherally (IP) at dose ranges of 2.5 to 10 mg/kg in rats and 3 to 30 mg/kg in mice. FLX (30 mg/kg) reduced binge eating, whereas peripherally administered psilocybin (3 mg/kg) was ineffective on reducing binge eating in male mice [90]. In male and female mice, the peripherally administered nociceptin receptor antagonist, SB612111(10 mg/kg), reduced binge food intake in rats with the intermittent access schedule, whereas FLX (30 mg/kg) had a feeding reduction not specific to the binge paradigm [89]. FLX (10 mg/kg), lorcaserin (6 mg/kg), and fenfluramine (3 mg/kg) reduced binge eating in wild-type (WT) mice, but all serotonin-acting compounds were ineffective in mice lacking the serotonin (5HT) 2C receptor (5HT 2C null) [94]. In another study, FLX was compared with topiramate and baclofen in binge eating male mice [89]. In this study, two doses of FLX (10 or 30 mg/kg), while only one dose of baclofen (3 mg/kg), were effective at reducing binge eating [89]. In female rats with a history calorie restriction and stress, FLX (3 mg/kg) and topiramate (60 mg/kg) reduced binge food intake, whereas sibutramine (1 or 3 mg/kg) reduced binge food intake in all groups [98]. In contrast, the benzodiazepine, midazolam (5 or 10 mg/kg), increased binge food intake, except in those rats with a history of calorie restriction and stress exposure [89]. In a study in male rats with either a fat-rich or carbohydrate-rich binge food, FLX (2.5–10 mg/kg) preferentially suppressed carbohydrate-rich binge food, whereas as naloxone (1 mg/kg) suppressed fat-rich binge food [91]. Naloxone (1 mg/kg) increased FLX-induced suppression of both types of binge foods [91].

#### 3.6.2. Lisdexamfetamine (LDX) Compared with Other Compounds

Three studies [93,95,97] compared LDX with other compounds. In all studies, LDX was administered orally at dose ranges of 0.15–1.5 mg/kg in mice and 0.1–100 mg/kg rats. Acute dosing with sibutramine (1 or 3 mg/kg) reduced binge eating; however, binge eating was not reduced at the 0.15–1.5 mg/kg dose range of LDX in female mice [93]. In female rats, oral dosing of a nootropic agent, piracetam (200 mg/kg), reduced binge eating to a similar extent as LDX (100 mg/kg) [95]. In a study comparing several compounds in female rats, nalmefene (0.1–1.0 mg/kg), SB-334867 (3–30 mg/kg), R-baclofen (1–10 mg/kg), sibutramine (1–5 mg/kg), rolipram (1–10 mg/kg), amphetamine (0.5–1.0 mg/kg), and LDX (0.3–1.5 mg/kg) reduced binge food intake. Sibutramine (0.3–5 mg/kg), rolipram (1–10 mg/kg), and baclofen (10 mg/kg) also reduced standard chow intake [97]. Prazosin (0.3 and 1.0 mg/kg) and SCH-23390 (0.1 mg/kg) attenuated LDX suppression of binge food intake [97].

**Table 5 biomolecules-13-00742-t005:** Combinational or direct comparisons of multiple compounds acting on different systems.

**Fluoxetine Compared with Other Compounds**			
**Study**	**Methods**	**Details**	**Major Findings**
Fadahunsi et al., 2022 [90]	**Species:** Mice **Binge Food:** High-fat diet (58% fat) **Access:** 48 h (initial binge)/24 h (subsequent binges). Binge asessed after 2.5 h. Control group had binge food ad libitum. **Frequency:** Once a week **Duration:** 3 weeks **Calorie Restriction:** None	Male C57BL/6J mice were exposed to the binge eating paradigm To determine the influence of the serotinergic psycheldelic, psilocybin, mice received an IP injection of vehicle (saline), FLX (30 mg/kg), or psilocybin (3 mg/kg) 30 min prior to a scheduled binge.	Intermittent binge food access produced a binge phenotype compared with continuous access. FLX reduced intermittent binge intake. Psilocybin had no effect on binge eating. Intermittent binge intake following psilocybin was not significantly different from vehicle.
Hardaway et al., 2016 [92]	**Species:** Mice **Binge Food:** High-fat diet (60% Fat) **Access:** 1 h **Frequency:** Daily **Duration:** >14 days **Calorie Restriction:** None	Male and female C57BL/6J mice were exposed to the binge eating paradigm, continuous access, or no access to the binge food. Binge food intake was assessed following nociceptin receptor antagonist (SB 612111, 0.1–10 mg/kg; IP) or FLX (30 mg/kg; IP) administration.	SB 612111 at 10 mg/kg reduced binge intake in male and females. FLX non-specifically reduced high-fat diet intake in acute, binge, and continuous access conditions.
Xu et al., 2016 [94]	**Species:** Mice **Binge Food:** High-fat diet (40% kcal from fat) **Access:** 24 h; binge intake measured 2.5 h **Frequency:** Once per week **Duration:** >7 weeks **Calorie Restriction:** None	Transgenic mice were compared with wild-type (WT) mice. All were on C57 Bl/6 background. 5-Hydroxytryptamine (5HT) 2C Receptors (2C-null) mice, 2C null in dopamine (DA) neurons (DA-2C-KO), or 2C expressed only in DA neurons (DA-2C-RE). Mice were also exposed to 2C agonist, lorcaserin (6 mg/kg; IP), and other 5HT agents (FLX; 10 mg/kg; IP or d-fenfluramine; 3 mg/kg; IP).	In WT, binge intake was reduced with FLX and d-fenfluramine but had no effect on binge intake in 2C-null mice. FLX and d-fenfluramine reduced binge intake in DA-2C-RE mice, similar to WT mice. Lorcaserin reduced binge intake in DA-2C-RE and WT mice only. Lorcaserin reduced refeeding response, following 24 h food deprivation, in WT and DA-2C-KO mice.
Czyzyk et al., 2010 [89]	**Species:** Mice **Binge Food:** High-energy diet (Teklad 95217) **Access:** 48 h (initial binge)/24 h (subsequent binges). Binge asessed after 2.5 and 24 h. Control group had binge food ad libitum. **Frequency:** Continuous or once per week (Intermittent). **Duration:** ~8 weeks **Calorie Restriction:** None	Male C57BL/6 mice were split into 3 groups: Chow, Continuous, or Intermittent. Pharmacological evaluation of binge-like eating behavior was performed using baclofen (0.3, 1, and 3 mg/kg), topiramate (10, 30, and 100 mg/kg), and FLX (3, 10, and 30 mg/kg).	Intermittent binge food access produced a robust binge eating phenotype at 2.5 and 24 h compared with continuous access. FLX at 10 mg/kg and 30 mg/kg reduced intermittent binge food intake at 2.5 h, only 30 mg/kg reduced 24 h binge food intake. Baclofen (3 mg/kg) reduced intermittent binge food intake at 2.5 h. Topiramate was not effective at reducing binge eating.
Cifani et al., 2009 [98]	**Species:** Rat **Binge food:** Paste mixture of Nutella (52%), food pellets (33%), and water (15%). **Access:** 2 h for days 5–6 and 13–14 of the 8-day cycles (second cycle). **Frequency:** Three 8-day cycle: 1–4 days of calorie restriction, 5–8 days ad libitum feeding. **Duration:** 24 days (test day 25). **Calorie Restriction:** 4 days per week at 66% restriction of ad libitum chow. **Stress:** 15 min exposure to sight and smell of binge food without the ability to consume it. (*Frustration Stress*)	Female Sprague Dawley rats were split into 4 groups: nonrestricted and not exposed to stress (NR + NS), restricted and not exposed to stress (R + NS), nonrestricted and exposed to stress (NR + S), and restricted and exposed to stress (R + S). Animals were exposed to the binge eating paradigm. The predictive validity of the restriction/refeeding/frustration stress model of binge eating was evaluated. The effects of FLX (3, 10 mg/kg), sibutramine (1, 3 mg/kg), topiramate (30, 60 mg/kg), and midazolam (5, 10 mg/kg) with vehicle were assessed.	Combination of cyclic R + S increased binge eating behavior. FLX at 3 mg/kg reduced binge food intake in the R + S group, whereas 10 mg/kg reduced binge food intake in all groups, except in the groups with cyclic restrict (R + NS). Sibutramine (1 and 3 mg/kg) reduced binge food intake in all groups. Topiramate (60 mg/kg) reduced binge eating in R + S group only. Midazolam (5 and 10 mg/kg) increased binge food intake in all groups, except in the R + S group.
Hagan et al., 1997 [91]	**Species:** Rat **Binge food:** Kellogg’s Froot Loops cereal (4.0 Kcal/g; 88% carbohydrate, 3% fat, 6% protein) given as palatable carbohydrate-rich food. Mars almond M&Ms (5.48 kCal/g; 57% carbohydrate, 30% fat, 8& protein) given as palatable fat-rich food. **Access:** 2 h **Frequency:** Daily (D) after injections for 2 h for each of the two experiments. **Duration:** 2 experiments in isolation 3 months apart; ~5 months total. **Calorie Restriction:** Animals were food-deprived 24 h before testing to induce feeding.	Male Sprague Dawley rats given ad libitum access to food and water. During eating tests, only the cereal and chocolate were available (no rat chow) along with water. Animals were randomly assigned to be pretreated with either SQ 1 mg/kg NAL or vehicle followed by an IP injection of one of 4 doses of FLX (0, 2.5, 5, and 10 mg/kg) with acess to binge food. Intakes were measured after 30 min and 2 h. At a 3 month time point, rats were pretreated with one of 4 doses of naloxone (NAL; 0, 0.01, 0.1, and 1 mg/kg); 10 min later, they were IP injected with 2.5 mg/kg FLX of vehicle and presented 20 min later with binge food access.	FLX at >2.5 mg/kg, alone, preferentially suppressed carbohydrate-rich binge food intake at 30 min and 2 h. NAL at 1 mg/kg naloxone, alone, preferentially suppressed fat-rich binge food intake at 30 min and 2 h. FLX (2.5 mg/kg) with NAL (1 mg/kg) reduced both type of binge foods. NAL (1 mg/kg) increased the FLX-induced suppression at every FLX dose level for both types of binge food.
**Lisdexamfetamine compared with other compounds**			
**Study**	**Methods**	**Details**	**Major Findings**
Sachdeo et al., 2019 [93]	**Species:** Mice **Binge Food:** Vegetable shortening blended with 10% sucrose **Access:** 30 min **Frequency:** 2 non- consecutive days per week **Duration:** 6 weeks **Calorie Restriction:** Two groups of mice had a 24 h calorie restriction prior to the binge period twice a week.	Female mice (C57Bl/6J background) with variant of *OPRM1* gene were exposed to binge eating paradigm to determine the polymorphisms’ (A112G) influence on binge-like eating. The interactions of polymorphisms, binge eating, and lisdexamfetamine (LDX: 0.15–1.5 mg/kg: PO) and sibutramine (SIB; 0.3–3 mg/kg;PO) were also examined.	No genotype differences in binge-like eating behavior. Acute LDX or SIB did not produce genotype-dependent differences binge-like eating. No genotype differences were reported in chronic (14-day) LDX or SIB.
Hussain & Krishnamurthy, 2018 [95]	**Species:** Rat **Binge Food:** Oreo cookies **Access:** 2 h **Frequency:** Alternate days **Duration:** 28 days **Calorie Restriction:** None **Stress:** Rats were single housed at time of binge (Isolation stress)	Female Wister albino rats were exposed to binge or binge + stress conditions. A nootropic agent, piracetam (200 mg/kg, oral) was compared with lisdexamfetamine (LDX; 100 mg/kg, oral) for binge eating suppression in the binge + stress conditions.	Piracetam had a similar profile in binge eating reduction to LDX. Piracetam as well as LDX reduced binge eating induced anxiety-like behaviors. Regional dopamine levels and vascular endothelial growth factor (*VEGF*) expression were similar to piracetam and LDX.
Vickers et al., 2015 [97]	**Species:** Rat **Binge Food:** Ground chocolate (Cadbury’s Dairy Milk) **Access:** 2 h **Frequency:** Intermittent on non-consecutive days, 3 days a week. **Duration:** 28 days **Calorie Restriction:** None	Female Wistar rats were exposed to a binge eating paradigm. The dose-dependent effects of several compounds were examined; these included opioid antagonist nalmefene (0.1–1.0 mg/kg, SQ), orexin antagonist SB-334867 (3.0–30 mg/kg, IP), antipsychotic olanzapine (0.3–3.0 mg/kg, IP), GABA_B_ agonist baclofen (1.0–10 mg/kg, IP), selective phosphodiesterase-4 inhibitor antidepressant rolipram (1.0–10 mg/kg, IP), norepinephrine and serotonin reuptake inhibitor sibutramine (0.3–5 mg/kg, PO), d-amphetamine (0.1–1.0 mg/kg, PO), and prodrug lisdexamfetamine (LDX; 0.1–1.5 mg/kg.,PO). To further investigate the effects of LDX, noradrenergic: prazosin (0.3 or 1.0 mg/kg, IP) or RX821002 (0.1 or 0.3 mg/kg, IP) or dopamine: raclopride (0.1 or 0.5 mg/kg, IP) or SCH-23390 (0.1 or 0.3 mg/kg, IP) in combination with LDX (1.0 mg/kg, PO) were considered.	Nalmefene (0.1–1.0 mg/kg), SB-334867 (3–30 mg/kg), R-baclofen (1–10 mg/kg), sibutramine (1–5 mg/kg), rolipram (1–10 mg/kg), amphetamine (0.5–1.0 mg/kg), and LDX (0.3–1.5 mg/kg) reduced binge food intake. Sibutramine (0.3–5 mg/kg) and baclofen (10 mg/kg) also reduced standard chow intake. Rolipram (1–10 mg/kg) reduced binge food and chow intake. Prazosin (0.3 and 1.0 mg/kg) attenuated LDX suppression of binge food. SCH-23390 (0.1 mg/kg) also attenuated the LDX binge food suppression.
**Additional compound comparisons or combinations**			
**Study**	**Methods**	**Details**	**Major Findings**
Price et al., 2018 [96]	**Species:** Rat **Binge Food:** High-fat diet (45%) **Access:** 2 h (after a single 7-day ad libitum access period) **Frequency:** Once per week **Duration:** 5–9 weeks **Calorie Restriction:** None	Adult male Sprague Dawley rats were SQ administered lorcaserin (5HT-2cR agonist; 0.25, 0.5, or 1.0 mg/Kg) or pimavanserin (5HT-2aR antagonist/inverse agonist; 0.3, 1.0, or 3.0 mg/kg). Binge eating intake or binge occurrence (intake over 2 h intake during day 6 binge food ad libitium period) was assessed.	Pimavanserin reduced binge intake over the range of doses tested with weight gain reductions at 1 and 3 mg/kg. Lorcaserin reduced binge intake at 1.0 mg/kg. Neither compound alone reduced binge occurrence. Combinations of pimavanserin (0.3 mg/kg) + lorcaserin (1.0 mg/kg) reduced binge occurrence, binge intake, and weight gain associated with binge food.
Corwin et al., 2016 [100]	**Species:** Rat **Binge Food:** Vegetable Shortening **Access:** 1 h **Frequency:** Intermittent on non-consecutive days or daily **Duration:** 5 to 8 weeks **Calorie Restriction:** None	Male Sprague Dawley rats were assessed for glutamate, GABA, dopamine, and opioid-related genes by qPCR 20 min before/after binge eating in PFC and VTA. Another set of rats were implanted with ventromedial (PL) or dorsomedial (M2) PFC cannula and injected with combined GABA R agonists (muscimol/baclofen), D1 R agonist (SKF 81297), D1 R antagonist (SCH 23390), D2 R agonist (quinpirole), and D2 R antagonist (eticlopride) before a scheduled binge episode.	In the VTA, *TH*, *D2R*, and *DAT* were higher pre-binge in the intermittent group, whereas *D1R* and *GABA_A_R* were deceased in the intermittent access group. In the PFC, *TH* and *GABA_B_R* were decreased in the intermittent access group, whereas *MOR-1* and *AMPA* were elevated in the post-binge conditions. In the intermittent access group, intra-PFC (M2 & PL) muscimol/baclofen increased binge intake. SKF 81297 or SCH 23390 did not influence binge intake. Quinpirole (10 µg) intra-PFC (M2) reduced binge intake in both access conditions. Eticlopride (0.3 µg) intra-PFC (M2) increased binge intake in the intermittent access group only.
Cao et al., 2014 [99]	**Species:** Mice **Binge Food:** High-fat diet (HFD; 40% fat) **Access:** 48 h initial exposure; 2.5 h **Frequency:** Once a week **Duration:** >2 weeks **Calorie Restriction:** None	Female C57BL/6 mice received bilateral ovariectomies (OVX) followed by 17β-estradiol (0.5 μg/d for 60 days; OVXE) or the containing vehicle (OVXV). Mice with tamoxifen-inducible CRE-recombinase on the tryptophan hydroxylase 2 (TPH2) promoter were used to visualize the serotonin-containing cells or to investigate the role of estrogen receptor alpha (ERα) on 5HT neurons. In addition, the roles of glucagon-like peptide (GLP-1) (4 ug/kg; SQ) as well as GLP-1 + E2 (4 ug/kg; SQ) were examined.	Binge food intake was higher in OVXV compared with OVXE, with no significant increase in body weight. KO of Erα in 5HT neurons reduced binge eating of OVXV to OVXE intake levels. An estrogen target gene, *Trim25*, was increased in OVXE and OVX + estrogen-GLP-1 (4 μg/kg) treatments. GLP-1 reduced binge food intake, but GLP-1-estrogen further reduced binge food intake in WT mice OVX. No effect of GLP-1 or GLP-1- estrogen was observed in Erα 5HT KO OVX mice.
Popik et al., 2010 [102]	**Species:** Rat **Binge food:** Lard **Access:** 2 h **Frequency:** Daily **Duration:** 6 weeks **Calorie Restriction:** None	Male Sprague Dawley rats were exposed to a daily limited-access binge eating paradigm for 6 weeks. After 3 weeks, animals were given either sibutramine, memantine, or MTEP for 12 days, and their effects on binge eating behavior were assessed. In a separate experiment, rats were given ad lib access to standard chow and water for 2 weeks, followed by assignment of 1 of the 3 drugs for 7 consecutive days. Following treatment, animals were given vehicle and food consumption was measured for 5 days.	Sibutramine (7.5 mg/kg PO) decreased binge food intake, but increased chow consumption. Sibutramine effects disappeared after discontinuation of treatment. NMDA receptor antagonist memantine (5 mg/kg IP) decreased binge food intake, but increased chow consumption. Memantine’s feeding suppression effects persisted after discontinuation of treatment. The mGluR5 antagonist MTEP (7.5 mg/kg IP) had a similar non-significant trend as memantine. Sibutramine and memantine reduced ad libitum chow intake during the treatment phase.
Pankevich et al., 2010 [101]	**Species:** Mice **Binge Food:** High-fat diet **Access:** Ad libitum access for 1 week (experiment 1). 1 h daily for stress experiment. **Frequency:** Ad libitum or daily (D) **Duration:** ~2 months **Calorie Restriction:** 75% of average caloric intake.	Male C57BL/6J mice were exposed to a model of moderate caloric restriction (CR; 10–15% weight loss), and physiological and behavioral stress measures were assessed. Changes in leptin, orexigenic hormones, melanin-concentrating hormone (MCH), and orexin levels due to stress and caloric restriction were also assessed, in addition to the effect of the MCH receptor-1 antagonist GSK-856464 (30 mg/kg; IP) and the SSRI citalopram (CIT; 20 mg/kg; IP) on binge feeding behavior. Tail suspension test on day 19 of restriction, restraint stress, and HPA axis stress response on day 21 were also performed. Chronic variable stress (CVS) was studied in a separate experiment (*restraint stress).*	Following 3 weeks of CR, mice exhibited significant increases in immobile time in the tail suspension test and stress-induced corticosterone levels. Similar outcomes were produced by high-fat diet feeding/withdrawal. Orexigenic hormones, MCH, and orexin were significantly elevated in response to high-fat diet only in mice with a history of CR. GSK-856464 reduced caloric intake compared with the vehicle and CIT in mice with a history of CR.
Wong et al. 2009 [104]	**Species:** Rat **Binge food:** Fat/sucrose mixture (FSM) (3.2%, 10%, or 32% sucrose in vegetable shortening). **Access:** 1 h **Frequency:** Daily (D) or Intermittent (INT), 3 days/week. **Duration:** ~6–8 months **Calorie Restriction:** None	Male Sprague Daley rats were grouped according to two schedules of access, D or INT, to an optional FSM. The effects of the opioid antagonist naltrexone (0.03, 0.1, 0.3, 1 mg/kg; IP), the dopamine-2-like (D2) antagonist raclopride (0.03, 0.1, 0.3 mg/kg; IP), and the GABAA agonist baclofen (0.6, 1, 1.8, 3.2 mg/kg; IP) were assessed.	INT 3.2% and 10% sucrose of the FSM produced binge-like eating. INT or D 32% sucrose did not promote a binge eating phenotype. Baclofen reduced 3.2% and 10% FSM in either INT or D access conditions. Naltrexone reduced FSM intake and either INT or D access conditions. Raclopride (0.3 mg/kg) reduced 3.2% and 10% FSM intake in the INT and D access conditions. Raclopride (0.1 mg/kg) increased 3.2% FSM with INT access.
Will et al., 2004 [103]	**Species:** Rat **Binge Food:** High-fat diet **Access:** 2 h daily (D) **Frequency:** Daily during training, then every other day for treatment. **Duration:** 4–6 weeks **Calorie Restriction:** None	Adult male Sprague Dawley rats were implanted with bilateral guide cannulas into the nucleus accumbens core and one of the following four structures: Basolateral amygdala, central nucleus of the amygdala, posterior mediobasal amygdala, or the posterior lateral striatum. GABA_A_ agonist, muscimol (20 ng/0.25 uL/side bilaterally), or saline were infused into the selected sites followed immediately by accumbens DAMGO (0.25 mg/0.5 uL/side bilaterally) or saline.	Intra-accumbens DAMGO robustly increased binge eating. Muscimol injected in either the basolateral or central nucleus of the amygdala blocked intra-accumbens DAMGO-induced binge eating. Muscimol was not effective when injected into either the posterior mediobasal amygdala or the posterior lateral striatum.

#### 3.6.3. Additional Compound Comparisons or Combinations

Seven studies [96,99,100,101,102,103,104] compared or examined the combinational effects with additional compound or agents. In female rats, two serotonergic compounds peripherally administered, lorcaserin (5HT 2c R agonist) or pimavanserin (5HT 2a antagonist/inverse agonist), were examined alone or combination in a binge paradigm to assess binge intake and binge episode occurrence (>binge food intake during intermittent vs. continuous access) [96]. Lorcaserin alone at 1.0 mg/kg and primavanserin alone at 1 or 3 mg/kg reduced binge intake, whereas the combination of lorcaserin (1 mg/kg) + primavanserin (0.3 mg/kg) reduced binge intake, binge occurrence, and weight gain [96]. Regional differences in the prefrontal cortex (PFC) of male rats to dopamine agonist/antagonists (SKF 81297, SCH 23390, quinpirole, or eticlopride) or GABA receptor-combined agonists (muscimol/baclofen) were assessed in male rats [100]. Intra-PFC (either pre-limbic or dorsomedial region) of the combined GABA agonists and dopamine (D2) receptor antagonist eticloprde (0.3 µg) into the intra-PFC dorsomedial region increased binge eating behavior [100]. In female OVX mice, combinations of GLP-4 and E2 reduced binge food intake; this effect was abolished in mice lacking estrogen receptor alpha (ER-α) on 5 HT neurons [99]. Peripheral administration of memantine (5 mg/kg) reduced binge food intake to a similar level as sibutramine (7.5 mg/kg) in male rats; however, memantine feeding suppression persisted post-treatment [102]. Peripheral administration of the MCH receptor-1 antagonist, GSK-856464 (30 mg/kg), suppressed binge intake compared with peripherally administered citalopram (20 mg/kg) in mice with a history of caloric restriction [101]. In male rats with intermittent access to a binge food of a fat mixture with different levels of sucrose sweetening (3.2–32%), peripheral administration of naltrexone (0.03–1 mg/kg), baclofen (0.6–3.2 mg/kg), and raclopride (0.03–0.3 mg/kg) were assessed [104]. All naltrexone doses tested reduced the fat/sugar mixtures (3.3–32%), whereas all baclofen doses reduced only 3.2% and 10% sugar/fat mixtures. Raclopride at 0.3 mg/kg reduced 3.2% and 10% sugar/fat mixtures; in contrast, at 0.1 mg/kg, raclopride increased 3.2% sugar/fat binge food intake [104]. In male rats, intra-accumbens DAMGO binge eating was blocked by microinjections of muscimol (20 ng/bilaterally) in either the basolateral or central nucleus of the amygdala [103].

## 4. Discussion

The purpose of this systematic review was to provide an overview of the existing literature on candidate pharmacotherapies or drug targets in rodent models of binge eating behaviors. Based on exclusionary/inclusionary criteria, we identified 67 peer-reviewed studies that used novel compounds or agents to influence binge eating behaviors in rats and mice. Not all studies were designed to test candidate pharmacotherapies. However, there were some similarities in the experimental design among the studies. For all studies, binge food was a highly palatable (e.g., sugar, fat, or combination) optional food item or diet. Each study employed an intermittent or limited access schedule of the binge food. In this sense, binge eating was related to the overconsumption of the highly palatable food, rather than overconsumption of standard (non-binge) diet. Binge eating behaviors in rodents are dependent on increasing the eating rate and amount of binge food consumed in a defined episode. As such, the episodic reduction in the *amount* of binge food is the major criterion used to determine the effectiveness of the potential pharmacotherapeutic agent. Using the amount of binge food consumed presents an inherent challenge when interpreting the findings from binge eating animal studies to potential therapeutic strategies for BED or BN, since clinical binge eating studies use a reduction in the *number of binge days* or binge episode *frequency* as one of the primary efficacy endpoints [22,105]. One way to address this challenge in rodent studies is to incorporate another dimension to binge eating measurements. For instance, *binge occurrence* (i.e., episodes of increase intake above baseline) or *binge propensity* (i.e., above the quartile of intake over episodes) are two measurements that could be incorporated into some binge eating paradigms to determine effectiveness of treatment [86,96,106]. Nonetheless, the measurements of binge amount, binge occurrence, and binge propensity do not fully encompass the interoceptive drive initiating binge eating bouts in persons with eating disorders.

Because FLX and LDX are FDA-approved medications for BN or BED [9,25], the incorporation of these medications in rodent binge eating studies assessing candidate pharmacotherapies should be given consideration. Of course, having a FLX or LDX treatment arm incorporated as a comparator is not feasible for all study designs, but would add relevance to findings. Indeed, 15 of the 67 studies included in this systematic review had used FLX or LDX. These studies represent a strong foundation for future studies of drug comparisons using rodent models.

## 5. Conclusions

Binge eating rodent models have provided valuable preclinical assessments for the screening and development of candidate pharmacotherapies. As with any animal model for a psychiatric condition, there are limitations in the data interpretation and extrapolation of the findings. Two ways to reduce the limitations and advance the development of future pharmacotherapies are to incorporate more binge eating dimension measurements, such as binge proneness and binge occurrence, and to use existing comparator treatments, such as LDX and FLX, in the study design.

## Figures and Tables

**Figure 1 biomolecules-13-00742-f001:**
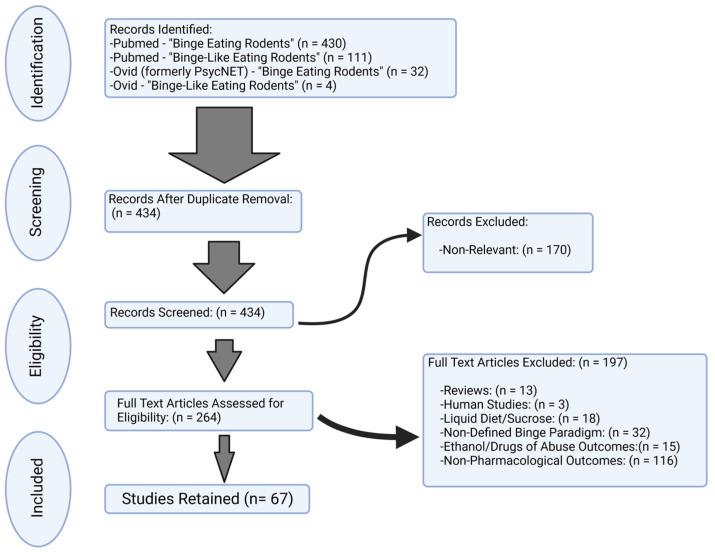
Flow chart for inclusion/exclusion of peer-reviewed studies. In accordance with Preferred Reporting Items for Systematic Reviews and Meta-Analyses (PRISMA) guidelines, a total of 67 studies were retained for the examination of pharmacological or compound assessment of binge-like eating behavior in rodents.

## Data Availability

Not applicable.

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
