# Peer review of "Systematic Review of Binge Eating Rodent Models for Developing Novel or Repurposing Existing Pharmacotherapies"

_biomolecules, 2023, doi:10.3390/biom13050742_

Round 1

Author Response

Reviewer #1

Berger and colleagues submit a Review of studies using rodents to simulate binge eating symptoms that included administration of biogenic amines, hormones, and other chemicals with potential pharmacological use in Binge Eating Disorder and Bulimia Nervosa. The ultimate goal was not very clear but it appears that they wanted to establish or at least suggest criteria for these type of studies with the aim of making them more relevant to criteria of effectiveness used in human binge eating pharmacological trials. Strengths were a comprehensive review of the studies (which yielded over 400 studies), logical reasoning behind exclusion of some of the studies, and a handy summary of the final 65 studies in table format listed alphabetically by types of biochemicals tested. Shortcomings include an inflated value of the two FDA approved drugs for binge eating and on the primary efficacy outcome reported in human studies, i.e., binge frequency. I have outlined some issues that if addressed, should render a more valuable and useful paper for animal and clinical researchers in this area.

Authors’ Response: We thank the reviewer for the careful consideration and review of the systematic review. You made some excellent points and we have attempted to address all your concerns in the revision. 

Major Issues

  1. The exact goal of the Review is not clear. The goal stated in the Abstract differs from that stated in the Introduction, lines 115-120. I suggest that authors go with the one stated in the Introduction for reasons described in the points below.

Authors’ Response: Yes, we have changed the abstract.

  1. The authors put too much stock in Prozac and Vivance. Their treatment efficacy is not impressive. These drugs add little to the benefits of CBT alone. Hence, I disagree that research with animal models of binge eating going forward should include arms testing these agents as “standard practice”. Research time and money is best spent testing novel therapeutics or novel serotonergic and dopaminergic ligands and mechanisms. Furthermore, the authors themselves provide a good bit of discussion about the dark side of these drugs (lines 75-85). Therefore, Lines 399-401 should be entirely omitted or at least tempered to suggest these additional arms be included but certainly not as “standard practice”.

Authors’ Response: Yes, we see your point. We have tempered our statement to now read “should be given consideration.”

  1. Similarly, the authors place too much value on the efficacy outcomes currently used in human binge eating studies, namely, frequency of binges. The entire clinical research process is under scrutiny (see Collins LM. Optimization of Behavioral, Biobehavioral, and Biomedical Interventions: The Multiphase Optimization Strategy (MOST). Springer International Publishing; 2018. doi:10.1007/978-3-319-72206-1). The authors themselves point out in lines 87-88 that “there is no guidance for industry on developing pharmacotherapies for the clinical management on binge eating”. Hence, it seems premature to call for animal research to simulate outcome measures used in human binge eating studies. The authors should not constrain themselves to frequency of binges as the outcome needing to be simulated in animals.. At the end, this is clearly what they appear to do. Instead, they should consider other measures indicative of improvement -without concern if they can be simulated in rats- such as duration of binges, size of binges, degree of sense of loss of control, craving, degree of caloric restriction, level of depression and anxiety. These are constructs that can and have been modeled in animals. For example, loss of control is indicated when rodents are willing to cross electric foot shock for food under sated conditions. Experience with bouts of caloric restriction have been associated with depression in rodents using the forced swim test. I understand that it would be difficult to include such animal studies in the paper if they did not include drug testing, but I suggest that the Discussion include a critical summation of human outcomes and consideration of testing for more than binge frequency. Lines 104-120 are in the right direction except for “loss of control” which should not be dismissed as unreplicable in animals. Engaging in animal/human dualism will only stunt progress in the treatment of eating disorders.

Authors Response: We agree on the complexity of huma studies and assessing efficacy for potential pharmacotherapies in rodents. We have included the reference from LM Collins for multiphase optimization strategies to suggest that a reduction binge frequency could be one component of effective innervation (line 97-98).

Yes, we have replaced “unreplicable in animals”, to read “difficult” and the sentence (110-112) to now read “While the sense of ‘loss of control’ of binge eating is a pathological feature that is difficult to objectively assessed in animals… “

  1. One issue I have with animal models of binge eating in general is whether the binge is truly different from the type of eating that would results from hunger (i.e., normal overeating). I am more convinced that true binge eating is occurring when a) caloric intake exceeds that of hungry rats (made hungry by mild restriction or ‘skipping a meal’), and b) when the binge includes proportionately more unhealthy (highly palatable) intake than healthier standard chow. Hungry rats will eat more chow than sated rats when given a mix of chow and palatable diet. It would be a valuable addition to the Table of studies to add if the study included a hungry control group (yes/no).

Authors response: Yes, we absolutely agree. Binge-type eating is defined differently by each investigator and rodent study. We have included “Calorie Restriction” in the Methods section for each study. In addition, we noted in the discussion that binge eating was in reference to the highly palatable food. “In this sense, binge eating was related to the overconsumption of the highly palatable food, rather than overconsumption of standard (non-binge) diet.” Line 392-394.

  1. Since the authors took pains to emphasize the need to simulate binge frequency in animal binge studies, they should describe how this would be accomplished. Vyvanse and Prozac do not work immediately in humans. Therefore, should rats be pre-treated daily with an equivalent dose of these drugs before any eating tests take place? Should rats be administered the experimental drug more than once to determine if the drugs continue to be effective over time? This assumes that the conditions that produce the binge eating in the rats would need to be maintained (as they are in the human condition). Etc...

Authors response: We understand your point. However, the length of treatment would be dependent on the study design. This would vary depending on expected outcomes.

  1. This Review paper’s value would be greatly increased if the authors ranked the animal studies in the Tables for veracity to human binge eating. This should be done not on the basis of having tested Vyvance or Prozac in the animals but on the conditions used to produce binge eating, and on the efficacy outcomes which should be efficacy outcomes important to clinical improvement. This could be done by adding an asterisk or other mark before the study in the existing tables.

Authors Response: We do not think it appropriate to rank the animal studies for relevance, since each rodent model should be specific for each hypothesis. That is, no animal encompasses the totality of complex psychiatric disease, such as binge eating disorder or bulimia nervosa. This is in accordance with NIH Notice of NIMH’s Considerations Regarding the Use of Animal Neurobehavioral Approaches in Basic and Pre-clinical Studies. 2019, NOT-MH-19-053. We have stated this point in the Introduction (line 105-108), which reads” In addition, animal models can assist in the early identification of some safety and tolerability-related issues. No animal model, however, can fully encompass all the intricacies of a psychiatric disorder. To be relevant and effective, animal models should be developed to ask specific hypothesis-testable questions for particular features of the complex psychiatric disease under investigation.”  

Writing Issues

  1. Abstract, lines 21-23, the aim of the study is not clear as written. I suggest the following, “Our purpose is to review the current literature to suggest criteria for determining pharmacological effectiveness in animal studies that are consistent with those used in determining effectiveness of pharmacological effectiveness in binge pathologies”. However, see point 1 above.

Authors Response: We have corrected this statement.

  1. Line 64, add “a” to “in a 26- week study”.

Authors Response: This has been added.

  1. Line 69, what is meant by “food-specific and non-planning impulsivity” in terms of BED

diagnostic criteria? If it is not relevant to BED criteria or features of BED, omit.

Authors Response: This has been deleted.

  1. Spell out PRISMA the first time it is used (Fig. 1 caption).

Authors Response: This has been added.

  1. Line 377, omit the word “examining”.

Authors Response: This has been deleted.

  1. Line 393, replace “or binge propensity” with “and binge propensity”.

Authors Response: This has been corrected.

  1. Line 412, replace “prone” with “proneness”.

Authors Response: This has been corrected. 

Reviewer 2 Report

This manuscript by Berger provided an overview of the potential pharmacotherapies or compounds tested in established rodent models of binge eating behavior.

Congratulations to authors, the review is very well written and of interest.

My suggestions are:

-       - First describe with details the Eating disorders in order to better introduce BED, in Introction section

-       - Use an acronym for Fluoxetine such ad FLX

-       - Use the acronyms all over the text

Author Response

This manuscript by Berger provided an overview of the potential pharmacotherapies or compounds tested in established rodent models of binge eating behavior.

Congratulations to authors, the review is very well written and of interest.

Authors response: Thank you for your review.

My suggestions are:

-       - First describe with details the Eating disorders in order to better introduce BED, in Introction section

Authors response: Yes, we have more information of BED and BN (lines 41-43)

-       Use an acronym for Fluoxetine such ad FLX

-     Use the acronyms all over the text
Authors response: FLX has been added.

Reviewer 3 Report

The systemic review provided by Berger et al is well conducted and complete. It presents an interesting and valuable overview on the studies evaluating the effects of pharmacological treatments on animal models of BED.

Nevertheless, I believe that the organization of the manuscript is not as appreciable. Indeed, a large amount of information is provided in tables only, while the results text is too limited.

Tables are a huge part of the manuscript. This is quite uncommon and difficult to read.

My indication is to reorganize the text so that most important aspects (e.g. experimental design, main results, species) are provided in the main text, while the tables recapitulate the most important info in a more schematized fashion. To this aim, tables should include more columns (species and sample sizes etc..).

Some minor errors in the writing can be easily found with a deeper evaluation, just a few examples:

L29 "on behavioral and psychotherapies"

L51 "Recent evident"

L55 "hyperactive"

Author Response

The systemic review provided by Berger et al is well conducted and complete. It presents an interesting and valuable overview on the studies evaluating the effects of pharmacological treatments on animal models of BED.

Nevertheless, I believe that the organization of the manuscript is not as appreciable. Indeed, a large amount of information is provided in tables only, while the results text is too limited. Tables are a huge part of the manuscript. This is quite uncommon and difficult to read. My indication is to reorganize the text so that most important aspects  tables recapitulate the most important info in a more schematized fashion. To this aim, tables should include more columns (species and sample sizes etc..).

Authors Response: We appreciate the reviewer’s comments. As you know, there is little consensus in the design and definition of binge eating in rodents among investigators. The tables were our way to organize the literature based on the methods (i.e., binge food, binge paradigm, species), design, and major findings. We have included this point in the Results sections (line 155-158), which reads:

 “These studies are summarized in Tables 1- 5. Given the differences in study design, approach, binge food, and major pharmacological outcomes the summary of these studies are best represented in a table format.”

Some minor errors in the writing can be easily found with a deeper evaluation, just a few examples:

L29 "on behavioral and psychotherapies"

Authors Response: Yes, we have added “cognitive” to “behavioral and psychotherapies”

L51 "Recent evident"

Authors Response: We have corrected “evident” to “evidence”

L55 "hyperactive"

Authors Response: We corrected this to hyperactivity.

Round 2

Reviewer 3 Report

Although the revised manuscript still presents a strong unbalance between main text and tables information, which is atypical, the authors provided argumentation to their style choice and fixed many minor errors.